# The GAN is dead; long live the GAN!
# A Modern Baseline GAN

**Yiwen Huang**
Brown University

**Aaron Gokaslan**
Cornell University

**Volodymyr Kuleshov**
Cornell University

**James Tompkin**
Brown University

## Abstract

There is a widely-spread claim that GANs are difficult to train, and GAN architectures in the literature are littered with empirical tricks. We provide evidence against this claim and build a modern GAN baseline in a more principled manner. First, we derive a well-behaved regularized relativistic GAN loss that addresses issues of mode dropping and non-convergence that were previously tackled via a bag of ad-hoc tricks. We analyze our loss mathematically and prove that it admits local convergence guarantees, unlike most existing relativistic losses. Second, this loss allows us to discard all ad-hoc tricks and replace outdated backbones used in common GANs with modern architectures. Using StyleGAN2 as an example, we present a roadmap of simplification and modernization that results in a new minimalist baseline—R3GAN ("Re-GAN"). Despite being simple, our approach surpasses StyleGAN2 on FFHQ, ImageNet, CIFAR, and Stacked MNIST datasets, and compares favorably against state-of-the-art GANs and diffusion models. Code: https://www.github.com/brownvc/R3GAN

## 1 Introduction

Generative adversarial networks (GANs) let us generate high-quality images in a single forward pass. However, the original objective in Goodfellow *et al*. [13], is notoriously difficult to optimize due to its minimax nature. This leads to a fear that training might diverge at any point due to instability, and a fear that generated images might lose diversity through mode collapse. While there has been progress in GAN objectives [14, 22, 81, 52, 64], practically, the effects of brittle losses are still regularly felt. This notoriety has had a lasting negative impact on GAN research.

A complementary issue—partly motivated by this instability—is that existing popular GAN backbones like StyleGAN [29, 31, 30, 32] use many poorly-understood empirical tricks with little theory. For instance, StyleGAN uses a gradient penalized non-saturating loss [52] to increase stability (affecting sample diversity), but then employs a minibatch standard deviation trick [28] to increase sample diversity. Without tricks, the StyleGAN backbone still resembles DCGAN [60] from 2015, yet it is still the common backbone of SOTA GANs such as GigaGAN [26] and StyleGAN-T [70]. Advances in GANs have been conservative compared to other generative models such as diffusion models [20, 78, 33, 34], where modern computer vision techniques such as multi-headed self attention [87] and backbones such as preactivated ResNet [17], U-Net [63] and vision transformers (ViTs) [9] are the norm. Given outdated backbones, it is not surprising that there is a widely-spread belief that GANs do not scale in terms of quantitative metrics like Frechet Inception Distance [19].

We reconsider this situation: we show that by combining progress in objectives into a regularized training loss, GANs gain improved training stability, which allows us to upgrade GANs with modern backbones. First, we propose a novel objective that augments the relativistic pairing GAN loss (RpGAN; [22]) with zero-centered gradient penalties [52, 64], improving stability [14, 64, 52]. We show mathematically that gradient-penalized RpGAN enjoys the same guarantee of local convergence as regularized classic GANs, and that removing our regularization scheme induces non-convergence.

38th Conference on Neural Information Processing Systems (NeurIPS 2024).

Once we have a well-behaved loss, none of the GAN tricks are necessary [28, 31], and we are free to engineer a modern SOTA backbone architecture. We strip StyleGAN of all its features, identify those that are essential, then borrow new architecture designs from modern ConvNets and transformers [48, 97]. Briefly, we find that proper ResNet design [17, 67], initialization [99], and resampling [29, 31, 32, 100] are important, along with grouped convolution [95, 5] and no normalization [31, 34, 14, 88, 4]. This leads to a design that is simpler than StyleGAN and improves FID performance for the same network capacity (2.75 vs. 3.78 on FFHQ-256).

In summary, our work first argues mathematically that GANs need not be tricky to train via an improved regularized loss. Then, it empirically develops a simple GAN baseline that, without any tricks, compares favorably by FID to StyleGAN [29, 31, 32], other SOTA GANs [3, 42, 94], and diffusion models [20, 78, 86] across FFHQ, ImageNet, CIFAR, and Stacked MNIST datasets.

## 2 Serving Two Masters: Stability and Diversity with RpGAN $+R_1 + R_2$

In defining a GAN objective, we tackle two challenges: stability and diversity. Some previous work deals with stability [29, 31, 32] and other previous work deals with mode collapse [22]. To make progress in both, we combine a stable method with a simple regularizer that is grounded by theory.

### 2.1 Traditional GAN

A traditional GAN [13, 57] is formulated as a minimax game between a discriminator (or critic) $D_\psi$ and a generator $G_\theta$. Given real data $x \sim p_\mathcal{D}$ and fake data $x \sim p_\theta$ produced by $G_\theta$, the most general form of a GAN is given by:

$$\mathcal{L}(\theta, \psi) = \mathbb{E}_{z \sim p_z} \left[ f \left( D_\psi(G_\theta(z)) \right) \right] + \mathbb{E}_{x \sim p_\mathcal{D}} \left[ f \left( -D_\psi(x) \right) \right] \tag{1}$$

where $G$ tries to minimize $\mathcal{L}$ while $D$ tries to maximize it. The choice of $f$ is flexible [50, 44]. In particular, $f(t) = -\log(1 + e^{-t})$ recovers the classic GAN by Goodfellow *et al.* [13]. For the rest of this work, this will be our choice of $f$ [57].

It has been shown that Equation 1 has convex properties when $p_\theta$ can be optimized directly [13, 81]. However, in practical implementations, the empirical GAN loss typically shifts fake samples beyond the decision boundary set by $D$, as opposed to directly updating the density function $p_\theta$. This deviation leads to a significantly more challenging problem, characterized by susceptibility to two prevalent failure scenarios: mode collapse/dropping[1] and non-convergence.

### 2.2 Relativistic $f$-GAN

We employ a slightly different minimax game named relativistic pairing GAN (RpGAN) by Jolicoeur-Martineau *et al.* [22] to address mode dropping. The general RpGAN is defined as:

$$\mathcal{L}(\theta, \psi) = \mathbb{E}_{\substack{z \sim p_z \\ x \sim p_\mathcal{D}}} \left[ f \left( D_\psi(G_\theta(z)) - D_\psi(x) \right) \right] \tag{2}$$

Although Eq. 2 differs only slightly from Eq. 1, evaluating this critic difference has a fundamental impact on the landscape of $\mathcal{L}$. Since Eq. 1 merely requires $D$ to separate real and fake data, in the scenario where all real and fake data can be separated by a single decision boundary, the empirical GAN loss encourages $G$ to simply move all fake samples barely past this single boundary—this degenerate solution is what we observe as mode collapse/dropping. Sun *et al.* [81] characterize such degenerate solutions as bad local minima in the landscape of $\mathcal{L}$, and show that Eq. 1 has *exponentially many* bad local minima. The culprit is the existence of a single decision boundary that naturally arises when real and fake data are considered in isolation. RpGAN introduces a simple solution by coupling real and fake data, *i.e.* a fake sample is critiqued by its realness *relative to* a real sample, which effectively maintains a decision boundary in the neighborhood of *each* real sample and hence forbids mode dropping. Sun *et al.* [81] show that the landscape of Eq. 2 contains no local minima that correspond to mode dropping solutions, and that every basin is a global minimum.

---

[1]While mode collapse and mode dropping are technically distinct issues, they are used interchangeably in this context to describe the common problem where $\text{supp}(p_\theta)$ does not comprehensively cover $\text{supp}(p_\mathcal{D})$. Mode collapse refers to the generator producing a limited diversity of samples (i.e., one image for the entire distribution), whereas mode dropping involves the generator failing to represent certain modes of the data distribution (ignoring entire subsets of the training distribution).

## 2.3 Training Dynamics of RpGAN

Although the RpGAN landscape result [81] allows us to address mode dropping, the training dynamics of RpGAN have yet to be studied. The ultimate goal of Eq. 2 is to find the equilibrium $(\theta^*, \psi^*)$ such that $p_{\theta^*} = p_\mathcal{D}$ and $D_{\psi^*}$ is constant everywhere on $p_\mathcal{D}$. Sun *et al.* [81] show that $\theta^*$ is globally reachable along a non-increasing trajectory in the landscape of Eq. 2 under reasonable assumptions. However, the existence of such a trajectory does not necessarily mean that gradient descent will find it. Jolicoeur-Martineau *et al.* show empirically that unregularized RpGAN does not perform well [22].

**Proposition I.** (Informal) *Unregularized RpGAN does not always converge using gradient descent.*

We confirm this proposition with a proof in Appendix B. We show analytically that RpGAN does not converge for certain types of $p_\mathcal{D}$, such as ones that approach a delta distribution. Thus, further regularization is necessary to fill in the missing piece of a well-behaved loss.

**Zero-centered gradient penalties.** To tackle RpGAN non-convergence, we explore gradient penalties as the solution since it is proven that zero-centered gradient penalties (0-GP) facilitate convergent training for classic GANs [52]. The two most commonly-used 0-GPs are $R_1$ and $R_2$:

$$R_1(\psi) = \frac{\gamma}{2}\mathbb{E}_{x \sim p_\mathcal{D}}\left[\|\nabla_x D_\psi\|^2\right]$$
$$R_2(\theta, \psi) = \frac{\gamma}{2}\mathbb{E}_{x \sim p_\theta}\left[\|\nabla_x D_\psi\|^2\right] \tag{3}$$

$R_1$ penalizes the gradient norm of $D$ on real data, and $R_2$ penalizes the gradient norm of $D$ on fake data. Analysis on the training dynamics of GANs has thus far focused on local convergence [55, 51, 52], *i.e.*, whether the training at least converges when $(\theta, \psi)$ are in a neighborhood of $(\theta^*, \psi^*)$. In such a scenario, the convergence behavior can be analyzed [55, 51, 52] by examining the spectrum of the Jacobian of the gradient vector field $(-\nabla_\theta \mathcal{L}, \nabla_\psi \mathcal{L})$ at $(\theta^*, \psi^*)$. The key insight here is that when $G$ already produces the true distribution, we want $\nabla_x D = 0$, so that $G$ is not pushed away from its optimal state, and thus the training does not oscillate. $R_1$ and $R_2$ impose such a constraint when $p_\theta = p_\mathcal{D}$. This also explains why earlier attempts at gradient penalties, such as the one-centered gradient penalty (1-GP) in WGAN-GP [14], fail to achieve convergent training [52] as they still encourage $D$ to have a non-zero slope when $G$ has reached optimality.

Since the same insight also applies to RpGAN, we extend our previous analysis and show that:

**Proposition II.** (Informal) *RpGAN with $R_1$ or $R_2$ regularization is locally convergent subject to similar assumptions as in* Mescheder *et al.* [52].

In Appendix C, our proof similarly analyzes the eigenvalues of the Jacobian of the regularized RpGAN gradient vector field at $(\theta^*, \psi^*)$. We show that all eigenvalues have a negative real part; thus, regularized RpGAN is convergent in a neighborhood of $(\theta^*, \psi^*)$ for small enough learning rates [52].

**Discussion.** Another line of work [64] links $R_1$ and $R_2$ to instance noise [75] as its analytical approximation. Roth et al. [64] showed that for the classic GAN [13] by Goodfellow *et al.*, $R_1$ approximates convolving $p_\mathcal{D}$ with the density function of $\mathcal{N}(0, \gamma I)$, up to additional weighting and a Laplacian error term. $R_2$ likewise approximates convolving $p_\theta$ with $\mathcal{N}(0, \gamma I)$ up to similar error terms. The Laplacian error terms from $R_1, R_2$ cancel when $D_\psi$ approaches $D_{\psi^*}$. We do not extend Roth *et al.*'s proof [64] to RpGAN; however, this approach might provide complimentary insights to our work, which follows the strategy of Mescheder *et al.* [52].

## 2.4 A Practical Demonstration

We experiment with how well-behaved our loss is on StackedMNIST [46] which consists of 1000 uniformly-distributed modes. The network is a small ResNet [17] for $G$ and $D$ without any normalization layers [21, 91, 1, 85]. Through the use of a pretrained MNIST classifier, we can explicitly measure how many modes of $p_\mathcal{D}$ are recovered by $p_\theta$. Furthermore, we can estimate the reverse KL divergence between the fake and real samples $D_{\text{KL}}(p_\theta \parallel p_\mathcal{D})$ via the KL divergence between the categorical distribution of $p_\theta$ and the true uniform distribution.

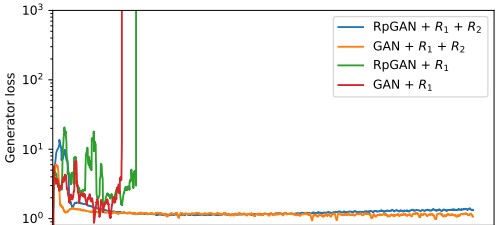

| Loss | # modes↑ | $D_{\text{KL}}$↓ |
|---|---|---|
| RpGAN $+R_1 + R_2$ | **1000** | **0.0781** |
| GAN $+R_1 + R_2$ | 693 | 0.9270 |
| RpGAN $+R_1$ | Fail | Fail |
| GAN $+R_1$ | Fail | Fail |

Figure 1: Generator $G$ loss for different objectives over training. Regardless of which objective is used, training diverges with only $R_1$ and succeeded with both $R_1$ and $R_2$. Convergence failure with only $R_1$ was noted by Lee et al. [42].

Table 1: StackedMNIST [46] result for each loss function. The maximum possible mode coverage is 1000. "Fail" indicates that training diverged early on.

A conventional GAN loss with $R_1$, as used by Mescheder et al. [52] and the StyleGAN series [29, 31, 32], diverges quickly (Fig. 1). Next, while theoretically sufficient for local convergence, RpGAN with only $R_1$ regularization is also unstable and diverges quickly[2]. In each case, the gradient of $D$ on fake samples explodes when training diverges. With both $R_1$ and $R_2$, training becomes stable for both the classic GAN and RpGAN. Now stable, we can see that the classic GAN suffers from mode dropping, whereas RpGAN achieves full mode coverage (Tab. 1) and reduces $D_{\text{KL}}$ from 0.9270 to 0.0781. As a point of contrast, StyleGAN [29, 31, 30, 32] uses the minibatch standard deviation trick to reduce mode dropping, improving mode coverage from 857 to 881 on StackedMNIST[3] and with barely any improvement on $D_{\text{KL}}$ [28].

$R_1$ alone is not sufficient for globally-convergent training. While a theoretical analysis of this is difficult, our small demonstration still provides insights into the assumptions of our convergence proof. In particular, the assumption that $(\theta, \psi)$ are sufficiently close to $(\theta^*, \psi^*)$ is highly unlikely early in training. In this scenario, if $D$ is sufficiently powerful, regularizing $D$ solely on real data is not likely to have much effect on $D$'s behavior on fake data and so training can fail due to an ill-behaved $D$ on fake data. This observation has been made by previous studies [84, 83] specifically for empirical GAN training, that regularizing an empirical discriminator with only $R_1$ leads to gradient explosion on fake data due to the memorization of real samples.

Thus, the practical solution is to regularize $D$ on both real and fake data. The benefit of doing so can be viewed from the insight of Roth *et al.* [64]: that applying $R_1$ and $R_2$ in conjunction smooths both $p_\mathcal{D}$ and $p_\theta$ which makes learning easier than only smoothing $p_\mathcal{D}$. We also find empirically that with both $R_1$ and $R_2$ in place, $D$ tends to satisfy $\mathbb{E}_{x \sim p_\mathcal{D}}\left[\|\nabla_x D\|^2\right] \approx \mathbb{E}_{x \sim p_\theta}\left[\|\nabla_x D\|^2\right]$ even early in the training. Jolicoeur-Martineau *et al.* [23] show that in this case $D$ becomes a maximum margin classifier—but if only one regularization term is applied, this does not hold. Additionally, having roughly the same gradient norm on real and fake data potentially reduces discriminator overfitting, as Fang *et al.* [10] observe that the gradient norm on real and fake data diverges when $D$ starts to overfit.

## 3 A Roadmap to a New Baseline — R3GAN

The well-behaved RpGAN + $R_1$ + $R_2$ loss alleviates GAN optimization problems, and lets us proceed to build a minimalist baseline—R3GAN—with recent network backbone advances in mind [48, 97]. Rather than simply state the new approach, we will draw out a roadmap from the StyleGAN2 baseline [30]. This model (Config A; identical to [30]) consists of a VGG-like [73] backbone for $G$, a ResNet $D$, a few techniques that facilitate style-based generation, and many tricks that serve as patches to the weak backbone. Then, we remove all non-essential features of StyleGAN2 (Config B), apply our loss function (Config C), and gradually modernize the network backbone (Config D-E).

---

[2] Varying $\gamma$ from 0.1 to 100 does not stabilize training.

[3] These numbers are from Karras *et al.* [28], Table 4. "857" corresponds to a low-capacity version of a progressive GAN and "881" adds the minibatch standard deviation trick. Further comparisons via loss curves are difficult since progressive GAN is a substantially different model than the small ResNet we use for this experiment.

We evaluate each configuration on FFHQ $256 \times 256$ [29]. Network capacity is kept roughly the same for all configurations—both $G$ and $D$ have about 25 M trainable parameters. Each configuration is trained until $D$ sees 5 M real images. We inherit training hyperparameters (*e.g.*, optimizer settings, batch size, EMA decay length) from Config A unless otherwise specified. We tune the training hyperparameters for our final model and show the converged result in Sec. 4.

**Minimum baseline (Config B).** We strip away all StyleGAN2 features, retaining only the raw network backbone and basic image generation capability. The features fall into three categories:

- Style-based generation: mapping network [29], style injection [29], weight modulation/demodulation [31], noise injection [29].
- Image manipulation enhancements: mixing regularization [29], path length regularization [31].
- Tricks: $z$ normalization [28], minibatch stddev [28], equalized learning rate [28], lazy regularization [31].

Following [69, 70], we reduce the dimension of $z$ to 64. The absence of equalized learning rate necessitates a lower learning rate, reduced from $2.5 \times 10^{-3}$ to $5 \times 10^{-5}$. Despite a higher FID of 12.46 than Config-A, this simplified baseline produces reasonable sample quality and stable training. We compare this with DCGAN [60], an early attempt at image generation. Key differences include:

| | Configuration | FID↓ | G #params | D #params |
|---|---|---|---|---|
| A | StyleGAN2 | 7.516 | 24.767M | 24.001M |
| B | Stripped StyleGAN2 | | | |
| | - $z$ normalization | | | |
| | - Minibatch stddev | | | |
| | - Equalized learning rate | | | |
| | - Mapping network | | | |
| | - Style injection | | | |
| | - Weight mod / demod | 12.46 | 18.890M | 23.996M |
| | - Noise injection | | | |
| | - Mixing regularization | | | |
| | - Path length regularization | | | |
| | - Lazy regularization | | | |
| C | Well-behaved Loss | | | |
| | + RpGAN loss | 11.77 | 18.890M | 23.996M |
| | + $R_2$ gradient penalty | 11.65 | | |
| D | ConvNeXt-ify pt. 1 | | | |
| | + ResNet-ify G & D | 10.17 | 23.400M | 23.282M |
| | - Output skips | 9.950 | 23.378M | |
| E | ConvNeXt-ify pt. 2 | | | |
| | + ResNeXt-ify G & D | 7.507 | 23.188M | 23.091M |
| | + Inverted bottleneck | 7.045 | 23.058M | 23.010M |

Table 2: Effect of our simplification and modernization efforts evaluted on FFHQ-256.

a) Convergent training objective with $R_1$ regularization.
b) Smaller learning rate, avoiding momentum optimizer (Adam $\beta_1 = 0$).
c) No normalization layer in $G$ or $D$.
d) Proper resampling via bilinear interpolation instead of strided (transposed) convolution.
e) Leaky ReLU in both $G$ and $D$, no tanh in the output layer of $G$.
f) $4 \times 4$ constant input for $G$, output skips for $G$, ResNet $D$.

**Experimental findings from StyleGAN.** Violating a), b), or c) often leads to training failures. Gidel *et al*. [11] show that *negative* momentum can improve GAN training dynamics. Since optimal negative momentum is another challenging hyperparameter, we do not use any momentum to avoid worsening GAN training dynamics. Studies suggest normalization layers harm generative models [31, 34]. Batch normalization [21] often cripples training due to dependencies across multiple samples, and is incompatible with $R_1$, $R_2$, or RpGAN that assume independent handling of each sample. Weaker data-independent normalizations [31, 34] might help; we leave this for future work. Early GANs may succeed despite violating a) and c), possibly constituting a full-rank solution [52] to Eq. 1.

Violations of d) or e) do not significantly impair training stability but negatively affect sample quality. Improper transposed convolution can cause checkerboard artifacts, unresolved even with subpixel convolution [72] or carefully tuned transposed convolution unless a low-pass filter is applied. Interpolation methods avoid this issue, varying from nearest neighbor [28] to Kaiser filters [32]. We use bilinear interpolation for simplicity. For activation functions, smooth approximations of (leaky) ReLU, such as Swish [61], GELU [18], and SMU [2], worsen FID. PReLU [15] marginally improves FID but increases VRAM usage, so we use leaky ReLU.

All subsequent configurations adhere to a) through e). Violation of f) is acceptable as it pertains to the network backbone of StyleGAN2 [31], modernized in Config D and E.

**Well-behaved loss function (Config C).** We use the loss function proposed in Section 2 and this reduces FID to 11.65. We hypothesize that the network backbone in Config B is the limiting factor.

**General network modernization (Config D).** First, we apply the 1-3-1 bottleneck ResNet architecture [16, 17] to both $G$ and $D$. This is the direct ancestor of all modern vision backbones [48, 97].

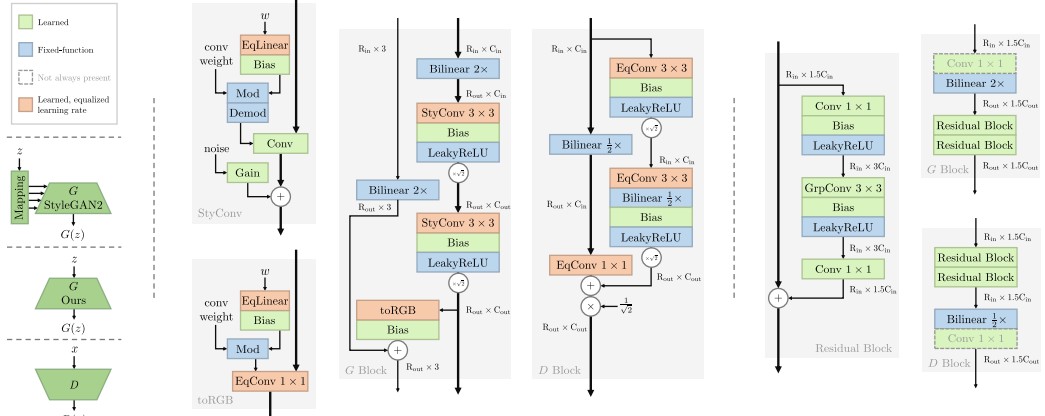

(a) Overall view      (b) StyleGAN2 architecture blocks [31] (Config A)      (c) Ours (Config E)

Figure 2: **Architecture comparison.** For image generation, $G$ and $D$ are often both deep ConvNets with either partially or fully symmetric architectures. **(a)** StyleGAN2 [31] $G$ uses a network to map noise vector $z$ to an intermediate style space $\mathcal{W}$. We use a traditional generator as style mapping is not necessary for a minimal working model. **(b)** StyleGAN2's building blocks have intricate layers but are themselves simple, with a ConvNet architecture from 2015 [38, 73, 16]. ResNet's identity mapping principle is also violated in the discriminator. **(c)** We remove tricks and modernize the architecture. Our design has clean layers with a more powerful ConvNet architecture.

We also incorporate principles discovered in Config B and various modernization efforts from ConvNeXt [48]. We categorize the roadmap of ConvNeXt as follows:

i. Consistently beneficial: i.1) increased width with depthwise convolution, i.2) inverted bottleneck, i.3) fewer activation functions, and i.4) separate resampling layers.

ii. Negligible performance gain: ii.1) large kernel depthwise conv. with fewer channels, ii.2) swap ReLU with GELU, ii.3) fewer normalization layers, and ii.4) swap batch norm. with layer norm.

iii. Irrelevant to our setting: iii.1) improved training recipe, iii.2) stage ratio, and iii.3) 'patchify' stem.

We aim to apply i) to our model, specifically i.3 and i.4 for the classic ResNet, while reserving i.1 and i.2 for Config E. Many aspects of ii) were introduced merely to mimic vision transformers [47, 9] without yielding significant improvements [48]. ii.3 and ii.4 are inapplicable due to our avoidance of normalization layers following principle c). ii.2 contradicts our finding that GELU deteriorates GAN performance, thus we use leaky ReLU per principle e). Liu *et al*. emphasize large conv. kernels (ii.1) [48], but this results in slightly worse performance compared to wider $3\times3$ conv. layers, so we do not adopt this ConvNeXt design choice.

**Neural network architecture details.** Given i.3, i.4, and principles c), d), and e), we can replace the StyleGAN2 backbone with a modernized ResNet. We use a fully symmetric design for $G$ and $D$ with 25 M parameters each, comparable to Config-A. The architecture is minimalist: each resolution stage has one transition layer and two residual blocks. The transition layer consists of bilinear resampling and an optional $1\times1$ conv. for changing spatial size and feature map channels. The residual block includes five operations: Conv$1\times1\to$ Leaky ReLU $\to$ Conv$3\times3\to$ Leaky ReLU $\to$ Conv$1\times1$, with the final Conv$1\times1$ having no bias term. For the $4\times4$ resolution stage, the transition layer is replaced by a basis layer for $G$ and a classifier head for $D$. The basis layer, similar to StyleGAN [29, 31], uses $4\times4$ learnable feature maps modulated by $z$ via a linear layer. The classifier head uses a global $4\times4$ depthwise conv. to remove spatial extent, followed by a linear layer to produce $D$'s output. We maintain the width ratio for each resolution stage as in Config A, making the stem width $3\times$ as wide due to the efficient $1\times1$ conv. The $3\times3$ conv. in the residual block has a compression ratio of 4, following [16, 17], making the bottleneck width $0.75\times$ as wide as Config A.

To avoid variance explosion due to the lack of normalization, we employ fix-up initialization [99]: We zero-initialize the last convolutional layer in each residual block and scale down the initialization of the other two convolutional layers in the block by $L^{-0.25}$, where $L$ is the number of residual blocks. We avoid other fix-up tricks, such as excessive bias terms and a learnable multiplier.

**Bottleneck modernization (Config E).** Now that we have settled on the overall architecture, we investigate how the residual block can be modernized, specifically i.1) and i.2). First, we explore i.1 and replace the 3×3 convolution in the residual block with a grouped convolution. We set the group size to 16 rather than 1 (*i.e.* depthwise convolution as in ConvNeXt) as depthwise convolution is highly inefficient on GPUs and is not much faster than using a larger group size. With grouped convolution, we can reduce the bottleneck compression ratio to two given the same model size. This increases the width of the bottleneck to 1.5× as wide as Config A. Finally, we notice that the compute cost of grouped convolution is negligible compared to 1×1 convolution, and so we seek to enhance the capacity of grouped convolution. We apply i.2), which inverts the bottleneck width and the stem width, and which doubles the width of grouped convolutions without any increase in model size. Figure 2 depicts our final design, which reflects modern CNN architectures.

## 4 Experiments Details

### 4.1 Roadmap Insights on FFHQ-256 [29]

As per Table 2, Config A (vanilla StyleGAN2) achieves an FID of 7.52 using the official implementation on FFHQ-256. Config B with all tricks removed achieves an FID of 12.46—performance drops as expected. Config C, with a well-behaved loss, achieves an FID of 11.65. But, now training is sufficiently stable to improve the architecture.

Config D, which improves $G$ and $D$ based on the classic ResNet and ConvNeXt findings, achieves an FID of 9.95. The output skips of the StyleGAN2 generator are no longer useful given our new architecture; including them produces a worse FID of 10.17. Karras *et al.* find that the benefit of output skips is mostly related to gradient magnitude dynamics [32], and this has been addressed by our ResNet architecture. For StyleGAN2, Karras *et al.* conclude that a ResNet architecture is harmful to $G$ [31], but this is not true in our case as their ResNet implementation is considerably different from ours: 1) Karras *et al.* use one 3-3 residual block for each resolution stage, while we have a separate transition layer and two 1-3-1 residual blocks; 2) i.3) and i.4) are violated as they do not have a linear residual block [67] and the transition layer is placed on the skip branch of the residual block rather than the stem; 3) the essential principle of ResNet [16]—identity mapping [17]—is violated as Karras *et al.* divide the output of the residual block by $\sqrt{2}$ to avoid variance explosion due to the absence of a proper initialization scheme.

For Config E, we conduct two experiments that ablate i.1 (increased width with depthwise conv.) and i.2 (an inverted bottleneck). We add GroupedConv and reduce the bottleneck compression ratio to two given the same model size. Each bottleneck is now 1.5× the width of Config A, and the FID drops to 7.51, surpassing the performance of StyleGAN2. By inverting the stem and the bottleneck dimensions to enhance the capacity of GroupedConv, our final model achieves an FID of 7.05, exceeding StyleGAN2.

### 4.2 Mode Recovery — StackedMNIST [53]

We repeat the earlier experiment in 1000-mode convergence on StackedMNIST (unconditional generation), but this time with our updated architecture and with comparisons to SOTA GANs and likelihood-based methods (Tab. 3, Fig. 5). One advantage brought up of likelihood-based models such as diffusion over GANs is that they achieve mode coverage [7]. We find that most GANs struggle to find all modes. But, PresGAN [8], DDGAN [94], and our approach are successful. Further, our method outperforms all other tested GAN models in term of KL divergence.

| Model | # modes↑ | $D_{\mathrm{KL}}$↓ |
|---|---|---|
| DCGAN [60] | 99 | 3.40 |
| VEEGAN [80] | 150 | 2.95 |
| WGAN-GP [14] | 959 | 0.73 |
| PacGAN [46] | 992 | 0.28 |
| StyleGAN2 [31] | 940 | 0.42 |
| PresGAN [8] | **1000** | 0.12 |
| Adv. DSM [24] | **1000** | 1.49 |
| VAEBM [93] | **1000** | 0.087 |
| DDGAN [94] | **1000** | 0.071 |
| MEG [39] | **1000** | 0.031 |
| Ours—Config E | **1000** | **0.029** |

Table 3: StackedMNIST 1000-mode coverage.

### 4.3 FID — FFHQ-256 [29] (Optimized)

We train Config E model until convergence and with optimized hyperparameters and training schedule on FFHQ at 256×256 (unconditional generation) (Tab. 4, Figs. 4 and 6). Please see our supplemental material for training details. Our model outperforms existing StyleGAN methods, plus four more

| Model | NFE↓ | FID↓ |
|---|---|---|
| StyleGAN2 [31] | 1 | 3.78 |
| StyleGAN3-T [32] | 1 | 4.81 |
| StyleGAN3-R [32] | 1 | 3.92 |
| LDM [62] | 200 | 4.98 |
| ADM (DDIM) [7, 49] | 500 | 8.41 |
| ADM (DPM-Solver) [7, 49] | 500 | 8.40 |
| Diffusion Autoencoder [59, 49] | 500 | 5.81 |
| Ours—Config E | 1 | 2.75 |
| *With ImageNet feature leakage [41]:* | | |
| PolyINR* [74] | 1 | 2.72 |
| StyleGAN-XL* [69] | 1 | 2.19 |
| StyleSAN-XL* [82] | 1 | 1.68 |

Table 4: FFHQ-256. * denotes models that leak ImageNet features.

| Model | NFE↓ | FID↓ |
|---|---|---|
| StyleGAN2 [31, 45] | 1 | 3.32 |
| MSG-GAN [27, 45] | 1 | 2.7 |
| Anycost GAN [45] | 1 | 2.52 |
| VE [78, 33] | 79 | 25.95 |
| VP [78, 33] | 79 | 3.39 |
| EDM [33] | 79 | 2.39 |
| Ours—Config E | 1 | 1.95 |

Table 5: FFHQ-64.

recent diffusion-based methods. On this common dataset experimental setting, many methods (not listed here) use the bCR [101] trick—this has only been shown to improve performance on FFHQ-256 (not even at different resolutions of FFHQ) [101, 98]. We do not use this trick.

## 4.4 FID — FFHQ-64 [33]

To compare with EDM [33] directly, we evaluate our model on FFHQ at 64×64 resolution. For this, we remove the two highest resolution stages of our 256×256 model, resulting in a generator that is less than half the number of parameters as EDM. Despite this, our model outperforms EDM on this dataset and needs one function evaluation only (Tab. 5).

## 4.5 FID — CIFAR-10 [37]

We train Config E model until convergence and with optimized hyperparameters and training schedule on CIFAR-10 (conditional generation) (Tab. 6, Fig. 8). Our method outperforms many other GANs by FID even though the model has relatively small capacity. For instance, StyleGAN-XL [69] has 18 M parameters in the generator and 125 M parameters in the discriminator, while our model has a 40 M parameters between the generator and discriminator combined (Fig. 3). Compared to diffusion models like LDM or ADM, GAN inference is significantly cheaper as it requires only one network function evaluation compared to the tens or hundreds of network function evaluations for diffusion models without distillation.

Many state-of-the-art GANs are derived from Projected GAN [68], including StyleGAN-XL [69] and the concurrent work of StyleSAN-XL [82]. These methods use a pre-trained ImageNet classifier in the discriminator. Prior work has shown that a pre-trained ImageNet discriminator can leak ImageNet features into the model [41], causing the model to perform better when evaluating on FID since it relies on a pre-trained ImageNet classifier for the loss. But, this does not improve results in perceptual studies [41]. Our model produces its low FID without any ImageNet pre-training.

| Model | NFE↓ | FID↓ |
|---|---|---|
| BigGAN [3] | 1 | 14.73 |
| TransGAN [87] | 1 | 9.26 |
| ViTGAN [42] | 1 | 6.66 |
| DDGAN [94] | 4 | 3.75 |
| Diffusion StyleGAN2 [90] | 1 | 3.19 |
| StyleGAN2 + ADA [30] | 1 | 2.42 |
| StyleGAN3-R + ADA [32, 25] | 1 | 10.83 |
| DDPM [20] | 1000 | 3.21 |
| DDIM [76] | 50 | 4.67 |
| VE [78, 33] | 35 | 3.11 |
| VP [78, 33] | 35 | 2.48 |
| Ours—Config E | 1 | 1.96 |
| *With ImageNet feature leakage [41]:* | | |
| StyleGAN-XL* [69] | 1 | 1.85 |

Table 6: CIFAR-10 performance.

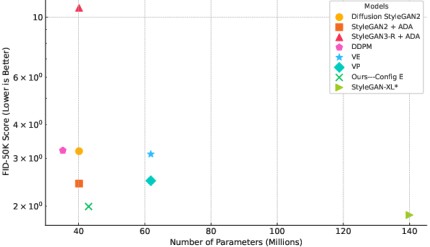

Figure 3: Millions of parameters vs. FID-50K (log scale) on CIFAR-10. Lower is better.

| Model | NFE↓ | FID↓ |
|---|---|---|
| DDPM++ [35] | 1000 | 8.42 |
| VDM [36] | 1000 | 7.41 |
| MSGAN [27, 56] | 1 | 12.3 |
| ADM [7] | 1000 | 3.60 |
| DDPM-IP [56] | 1000 | 2.87 |
| Ours—Config E | 1 | 1.27 |
| *With ImageNet feature leakage [41]:* | | |
| StyleGAN-XL* [69] | 1 | 1.10 |

Table 7: ImageNet-32.

| Model | NFE↓ | FID↓ |
|---|---|---|
| BigGAN-deep [3] | 1 | 4.06 |
| DDPM [20] | 250 | 11.0 |
| DDIM [76] | 50 | 13.7 |
| ADM [7] | $^\S$250 | 2.91 |
| EDM [33] | 79 | 2.23 |
| CT [79] | 2 | 11.1 |
| CD [79] | 3 | 4.32 |
| iCT-deep [77] | 2 | 2.77 |
| DMD [96] | 1 | 2.62 |
| Ours—Config E | 1 | 2.09 |
| *With ImageNet feature leakage [41]:* | | |
| StyleGAN-XL* [69] | 1 | 1.52 |

Table 8: ImageNet-64. $^\S$deterministic sampling.

## 4.6 FID — ImageNet-32 [6]

We train Config E model until convergence and with optimized hyperparameters and training schedule on ImageNet-32 (conditional generation). We compare against recent GAN models and recent diffusion models in Table 7. We adjust the number of parameters in the generator of our model to match StyleGAN-XL [69]'s generator (84M parameters). Specifically, we make the model significantly wider to match. Our method achieves comparable FID despite using a 60% smaller discriminator (Tab. 7) and despite not using a pre-trained ImageNet classifier.

## 4.7 FID — ImageNet-64 [6]

We evaluate our model on ImageNet-64 to test its scalability. We stack another resolution stage on our ImageNet-32 model, resulting in a generator of 104 M parameters. This model is nearly $3\times$ smaller than diffusion-like models [7, 33, 79, 77] that rely on the ADM backbone, which contains about 300 M parameters. Despite the smaller model size and that our model generates samples in one step, it outperforms larger diffusion models with many NFEs on FID (Tab. 8).

## 4.8 Recall

We evaluate the recall [40] of our model on each dataset to quantify sample diversity. In general, our model achieves a recall that is similar to or marginally worse than the diffusion model counterpart, yet superior to existing GAN models. For CIFAR-10, the recall of our model peaked at 0.57; as a point of comparison, StyleGAN-XL [69] has a worse recall of 0.47 despite its lower FID. For FFHQ, we obtain a recall of 0.53 at 64×64 and 0.49 at 256×256, whereas StyleGAN2 [31] achieved a recall of 0.43 on FFHQ-256. Our ImageNet-32 model achieved a recall of 0.63; comparable to ADM [7]. Our ImageNet-64 model achieved recall 0.59. While this is slightly worse than $\approx$0.63 that many diffusion models achieve, it is better than BigGAN-deep [3] which achieved a recall of 0.48.

## 5  Discussion and Limitations

We have shown that a simplification of GANs is possible for image generation tasks, built upon a more stable RpGAN+$R_1$ + $R_2$ objective with mathematically-demonstrated convergence properties that still provides diverse output. This stability is what lets us re-engineer a modern network architecture without the tricks of previous methods, producing the R3GAN model with competitive FID on the common datasets of Stacked-MNIST, FFHQ, CIFAR-10, and ImageNet as an empirical demonstration of the mathematical benefits.

The focus of our work is to elucidate the essential components of a minimum GAN for image generation. As such, we prioritize simplicity over functionality—we do not claim to beat the performance of every existing model on every dataset or task; merely to provide a new simple

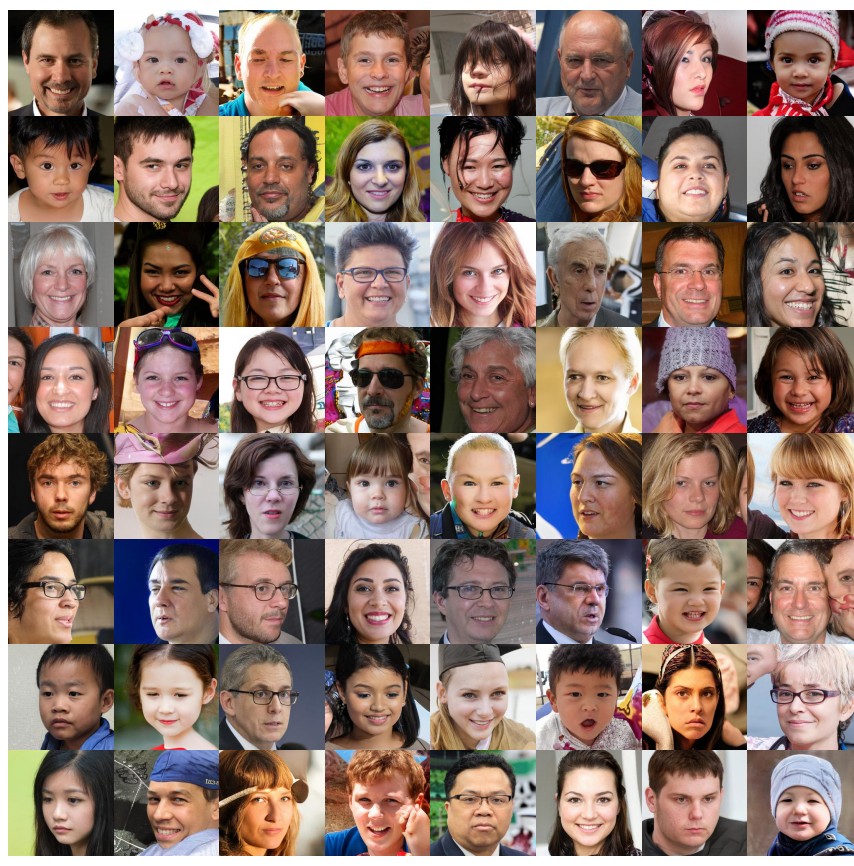

Figure 4: Qualitative examples of sample generation from our Config E on FFHQ-256.

baseline that converges easily. While this makes our model a possible backbone for future GANs, it also means that it is not suitable to apply our model directly to downstream applications such as image editing or controllable generation, as our model lacks dedicated features for easy image inversion or disentangled image synthesis. For instance, we remove style injection functionality from StyleGAN even though this has a clear use. We also omitted common techniques that have been shown in previous literature to improve FID considerably. Examples include some form of adaptive normalization modulated by the latent code [7, 33, 29, 98, 58, 89, 66], and using multiheaded self attention at lower resolution stages [7, 33, 34]. We aim to explore these techniques in a future study.

Further, our work is limited in its evaluation of the scalability of R3GAN models. While they show promising results on 64×64 ImageNet, we are yet to verify the scalability on higher resolution ImageNet data or large-scale text to image generation tasks [12].

Finally, as a method that can improve the quality of generative models, it would be amiss not to mention that generative models—especially of people—can cause direct harm (e.g., through personalized deep fakes) and societal harm through the spread of disinformation (e.g., fake influencers).

## 6   Conclusion

This work introduced R3GAN, a new baseline GAN that features increased stability, leverages modern architectures, and does not require ad-hoc tricks that are commonplace in existing GAN models. Central to our approach is a regularized relativistic loss that provably features local convergence and that improves the stability of GAN training. This stable loss enables us to ablate various tricks that were previously necessary in GANs, and incorporate in their place modern deep architectures. The resulting streamlined baseline achieves competitive performance to SOTA models within its parameter size class. We anticipate that our backbone will help to drive future GAN research.

**Acknowledgements.** The authors thank Xinjie Jayden Yi for contributing to the proof and Yu Cheng for helpful discussion. For compute, the authors thank Databricks Mosaic Research. Yiwen Huang was supported by a Brown University Division of Research Seed Award, and James Tompkin was supported by NSF CAREER 2144956. Volodymyr Kuleshov was supported by NSF CAREER 2145577 and NIH MIRA 1R35GM15124301.

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

# Appendices

## A Local convergence

Following [52], GAN training can be formulated as a dynamical system where the update operator is given by $F_h(\theta, \psi) = (\theta, \psi) + hv(\theta, \psi)$. $h$ is the learning rate and $v$ denotes the gradient vector field:

$$v(\theta, \psi) = \begin{pmatrix} -\nabla_\theta \mathcal{L}(\theta, \psi) \\ \nabla_\psi \mathcal{L}(\theta, \psi) \end{pmatrix} \tag{4}$$

Mescheder et al. [51] showed that local convergence near $(\theta^*, \psi^*)$ can be analyzed by examining the spectrum of the Jacobian $\mathbf{J}_{F_h}$ at the equilibrium: if the Jacobian has eigenvalues with absolute value bigger than 1, then training does not converge. On the other hand, if all eigenvalues have absolute value smaller than 1, then training will converge to $(\theta^*, \psi^*)$ at a linear rate. If all eigenvalues have absolute value equal to 1, the convergence behavior is undetermined.

Given some calculations [52], we can show that the eigenvalues of the Jacobian of the update operator $\lambda_{\mathbf{J}_{F_h}}$ can be determined by $\lambda_{\mathbf{J}_v}$:

$$\lambda_{\mathbf{J}_{F_h}} = 1 + h\lambda_{\mathbf{J}_v} . \tag{5}$$

That is, given small enough $h$ [52], the training dynamics can instead be examined using $\lambda_{\mathbf{J}_v}$, *i.e.*, the eigenvalues of the Jacobian of the gradient vector field. If all $\lambda_{\mathbf{J}_v}$ have a negative real part, the training will locally converge to $(\theta^*, \psi^*)$ at a linear rate. On the other hand, if some $\lambda_{\mathbf{J}_v}$ have a positive real part, the training is not convergent. If all $\lambda_{\mathbf{J}_v}$ have a zero real part, the convergence behavior is inconclusive.

## B DiracRpGAN: A demonstration of non-convergence

**Summary.** To obtain DiracRpGAN, we apply Eq. 2 to the DiracGAN [52] problem setting. After simplification, DiracRpGAN and DiracGAN are different only by a constant. They have the same gradient vector field, therefore all proofs are identical to Mescheder *et al*. [52].

**Definition B.1.** *The DiracRpGAN consists of a (univariate) generator distribution $p_\theta = \delta_\theta$ and a linear discriminator $D_\psi(x) = \psi \cdot x$. The true data distribution $p_\mathcal{D}$ is given by a Dirac distribution concentrated at 0.*

In this setup, the RpGAN training objective is given by:

$$\mathcal{L}(\theta, \psi) = f(\psi\theta) . \tag{6}$$

We can now show analytically that DiracRpGAN does not converge without regularzation.

**Lemma B.2.** *The unique equilibrium point of the training objective in Eq. 6 is given by $\theta = \psi = 0$. Moreover, the Jacobian of the gradient vector field at the equilibrium point has the two eigenvalues $\pm f'(0)i$ which are both on the imaginary axis.*

The gradient vector field $v$ of Eq. 6 is given by:

$$v(\theta, \psi) = \begin{pmatrix} -\nabla_\theta \mathcal{L}(\theta, \psi) \\ \nabla_\psi \mathcal{L}(\theta, \psi) \end{pmatrix} = \begin{pmatrix} -\psi f'(\psi\theta) \\ \theta f'(\psi\theta) \end{pmatrix} \tag{7}$$

and the Jacobian of $v$:

$$\mathbf{J}_v = \begin{pmatrix} -\psi^2 f''(\psi\theta) & -f'(\psi\theta) - \psi\theta f''(\psi\theta) \\ f'(\psi\theta) + \psi\theta f''(\psi\theta) & \theta^2 f''(\psi\theta) \end{pmatrix} \tag{8}$$

Evaluating $\mathbf{J}_v$ at the equilibrium point $\theta = \psi = 0$ gives us:

$$\mathbf{J}_v\Big|_{(0,0)} = \begin{pmatrix} 0 & -f'(0) \\ f'(0) & 0 \end{pmatrix} \tag{9}$$

Therefore, the eigenvalues of $\mathbf{J}_v$ are $\lambda_{1/2} = \pm f'(0)i$, both of which have a real part of 0. Thus, the convergence of DiracRpGAN is inconclusive and further analysis is required.

**Lemma B.3.** *The integral curves of the gradient vector field $v(\theta, \psi)$ do not converge to the equilibrium point. More specifically, every integral curve $(\theta(t), \psi(t))$ of the gradient vector field $v(\theta, \psi)$ satisfies $\theta(t)^2 + \psi(t)^2 = const$ for all $t \in [0, \infty)$.*

Let $R(\theta, \psi) = \frac{1}{2}(\theta^2 + \psi^2)$, then:

$$\frac{\mathrm{d}}{\mathrm{d}t} R(\theta(t), \psi(t)) = -\theta(t)\psi(t)f'(\theta(t)\psi(t)) + \psi(t)\theta(t)f'(\theta(t)\psi(t))$$
$$= 0 . \tag{10}$$

We see that the distance between $(\theta, \psi)$ and the equilibrium point $(0, 0)$ stays constant. Therefore, training runs in circles and never converges.

Next, we investigate the convergence behavior of DiracRpGAN with regularization. For DiracRpGAN, both $R_1$ and $R_2$ can be reduced to the following form:

$$R(\psi) = \frac{\gamma}{2}\psi^2 \tag{11}$$

**Lemma B.4.** *The eigenvalues of the Jacobian of the gradient vector field for the gradient-regularized DiracRpGAN at the equilibrium point are given by*

$$\lambda_{1/2} = -\frac{\gamma}{2} \pm \sqrt{\frac{\gamma^2}{4} - f'(0)} \tag{12}$$

*In particular, for $\gamma > 0$ all eigenvalues have a negative real part. Hence, gradient descent is locally convergent for small enough learning rates.*

With regularization, the gradient vector field becomes

$$\tilde{v}(\theta, \psi) = \begin{pmatrix} -\psi f'(\psi\theta) \\ \theta f'(\psi\theta) - \gamma\psi \end{pmatrix} \tag{13}$$

the Jacobian of $\tilde{v}$ is then given by

$$\mathbf{J}_{\tilde{v}} = \begin{pmatrix} -\psi^2 f''(\psi\theta) & -f'(\psi\theta) - \psi\theta f''(\psi\theta) \\ f'(\psi\theta) + \psi\theta f''(\psi\theta) & \theta^2 f''(\psi\theta) - \gamma \end{pmatrix} \tag{14}$$

evaluating the Jacobian at $\theta = \psi = 0$ yields

$$\mathbf{J}_{\tilde{v}}\Big|_{(0,0)} = \begin{pmatrix} 0 & -f'(0) \\ f'(0) & -\gamma \end{pmatrix} \tag{15}$$

given some calculations, we arrive at Eq.12.

## C    General Convergence Results

**Summary.**    The proofs are largely the same as Mescheder *et al*. [52]. We use the same proving techniques, and only slightly modify the assumptions and proof details to adapt Mescheder *et al*.'s effort to RpGAN. Like in [52], our proofs do not rely on unrealistic assumptions such as $\mathrm{supp}\, p_{\mathcal{D}} = \mathrm{supp}\, p_{\theta}$.

### C.1    Assumptions

We closely follow [52] but modify the assumptions wherever necessary to tailor the proofs for RpGAN. Like in [52], we also consider the realizable case where there exists $\theta$ such that $G_\theta$ produces the true data distribution.

**Assumption I.**    *We have $p_{\theta^*} = p_{\mathcal{D}}$, and $D_{\psi^*} = C$ in some local neighborhood of $\mathrm{supp}\, p_{\mathcal{D}}$, where $C$ is some arbitrary constant.*

Since RpGAN is defined on critic difference rather than raw logits, we no longer require $D_{\psi^*}$ to produce 0 on $\mathrm{supp}\, p_{\mathcal{D}}$, instead any constant $C$ would suffice.

**Assumption II.** *We have $f'(0) \neq 0$ and $f''(0) < 0$.*

This assumption is the same as in [52]. The choice $f(t) = -\log(1 + e^{-t})$ adopted in the main text satisfies this assumption.

As discussed in [52], there generally is not a single equilibrium point $(\theta^*, \psi^*)$, but a submanifold of equivalent equilibria corresponding to different parameterizations of the same function. It is therefore necessary to represent the equilibrium as *reparameterization manifolds* $\mathcal{M}_G$ and $\mathcal{M}_D$. We modify the reparameterization $h$ as follows:

$$h(\psi) = \mathbb{E}_{\substack{x \sim p_\mathcal{D} \\ y \sim p_\mathcal{D}}} \left[ |D_\psi(x) - D_\psi(y)|^2 + \|\nabla_x D_\psi(x)\|^2 \right] \tag{16}$$

to account for the fact that $D_{\psi^*}$ is now allowed to have any constant value on $\operatorname{supp} p_\mathcal{D}$. The *reparameterization manifolds* are then given by:

$$\mathcal{M}_G = \{\theta \,|\, p_\theta = p_\mathcal{D}\} \tag{17}$$
$$\mathcal{M}_D = \{\psi \,|\, h(\psi) = 0\} \tag{18}$$

We assume the same regularity properties as in [52] for $\mathcal{M}_G$ and $\mathcal{M}_D$ near the equilibrium. To state these assumptions, we need:

$$g(\theta) = \mathbb{E}_{x \sim p_\theta} \left[ \nabla_\psi D_\psi|_{\psi=\psi^*} \right] \tag{19}$$

which leads to:

**Assumption III.** *There are $\epsilon$-balls $B_\epsilon(\theta^*)$ and $B_\epsilon(\psi^*)$ around $\theta^*$ and $\psi^*$ so that $\mathcal{M}_G \cap B_\epsilon(\theta^*)$ and $\mathcal{M}_D \cap B_\epsilon(\psi^*)$ define $\mathcal{C}^1$-manifolds. Moreover, the following holds:*

(i) *if $v \in \mathbb{R}^n$ is not in $\mathcal{T}_{\psi^*}\mathcal{M}_D$, then $\partial_v^2 h(\psi^*) \neq 0$.*
(ii) *if $w \in \mathbb{R}^m$ is not in $\mathcal{T}_{\theta^*}\mathcal{M}_G$, then $\partial_w g(\theta^*) \neq 0$.*

These two conditions have exactly the same meanings as in [52]: the first condition indicates the geometry of $\mathcal{M}_D$ can be locally described by the second derivative of $h$. The second condition implies that $D$ is strong enough that it can detect any deviation from the equilibrium generator distribution. This is the only assumption we have about the expressiveness of $D$.

## C.2 Convergence

We can now show the general convergence result for gradient penalized RpGAN, consider the gradient vector field with either $R_1$ or $R_2$ regularization:

$$\tilde{v}_i(\theta, \psi) = \begin{pmatrix} -\nabla_\theta \mathcal{L}(\theta, \psi) \\ \nabla_\psi \mathcal{L}(\theta, \psi) - \nabla_\psi R_i(\theta, \psi) \end{pmatrix} \tag{20}$$

note that the convergence result can also be trivially extended to the case where both $R_1$ and $R_2$ are applied. We omit the proof for this case as it is redundant once the convergence with either regularization is proven.

**Theorem.** *Assume Assumption I, II and III hold for $(\theta^*, \psi^*)$. For small enough learning rates, gradient descent for $\tilde{v}_1$ and $\tilde{v}_2$ are both convergent to $\mathcal{M}_G \times \mathcal{M}_D$ in a neighborhood of $(\theta^*, \psi^*)$. Moreover, the rate of convergence is at least linear.*

We extend the convergence proof by Mescheder *et al.* [52] to our setting. We first prove lemmas necessary to our main proof.

**Lemma C.2.1.** *Assume $J \in \mathbb{R}^{(n+m)\times(n+m)}$ is of the following form:*

$$J = \begin{pmatrix} 0 & -B^\top \\ B & -Q \end{pmatrix} \tag{21}$$

*where $Q \in \mathbb{R}^{m\times m}$ is a symmetric positive definite matrix and $B \in \mathbb{R}^{m\times n}$ has full column rank. Then all eigenvalues $\lambda$ of $J$ satisfy $\Re(\lambda) < 0$.*

*Proof.* See Mescheder *et al.* [52], Theorem A.7.

**Lemma C.2.2.** *The gradient of $\mathcal{L}(\theta, \psi)$ w.r.t. $\theta$ and $\psi$ are given by:*

$$\nabla_\theta \mathcal{L}(\theta, \psi) = \mathbb{E}_{\substack{z \sim p_z \\ x \sim p_\mathcal{D}}} [f'(D_\psi(G_\theta(z)) - D_\psi(x)) \left[\nabla_\theta G_\theta(z)\right]^\top \nabla_x D_\psi(G_\theta(z))] \tag{22}$$

$$\nabla_\psi \mathcal{L}(\theta, \psi) = \mathbb{E}_{\substack{z \sim p_z \\ x \sim p_\mathcal{D}}} [f'(D_\psi(G_\theta(z)) - D_\psi(x))(\nabla_\psi D_\psi(G_\theta(z)) - \nabla_\psi D_\psi(x))] \tag{23}$$

*Proof.* This is just the chain rule.

**Lemma C.2.3.** *Assume that $(\theta^*, \psi^*)$ satisfies Assumption I. The Jacobian of the gradient vector field $v(\theta, \psi)$ at $(\theta^*, \psi^*)$ is then*

$$\mathbf{J}_v\Big|_{(\theta^*, \psi^*)} = \begin{pmatrix} 0 & -K_{DG}^\top \\ K_{DG} & K_{DD} \end{pmatrix} \tag{24}$$

*the terms $K_{DD}$ and $K_{DG}$ are given by*

$$K_{DD} = f''(0)\mathbb{E}_{\substack{x \sim p_\mathcal{D} \\ y \sim p_\mathcal{D}}}[(\nabla_\psi D_{\psi^*}(x) - \nabla_\psi D_{\psi^*}(y))(\nabla_\psi D_{\psi^*}(x) - \nabla_\psi D_{\psi^*}(y))^\top] \tag{25}$$

$$K_{DG} = f'(0)\nabla_\theta \mathbb{E}_{x \sim p_\theta}[\nabla_\psi D_{\psi^*}(x)]|_{\theta = \theta^*} \tag{26}$$

*Proof.* Note that

$$\mathbf{J}_v\Big|_{(\theta^*, \psi^*)} = \begin{pmatrix} -\nabla_\theta^2 \mathcal{L}(\theta^*, \psi^*) & -\nabla_{\theta, \psi}^2 \mathcal{L}(\theta^*, \psi^*) \\ \nabla_{\theta, \psi}^2 \mathcal{L}(\theta^*, \psi^*) & \nabla_\psi^2 \mathcal{L}(\theta^*, \psi^*) \end{pmatrix} \tag{27}$$

By Assumption I, $D_{\psi^*} = C$ in some neighborhood of $\text{supp } p_\mathcal{D}$. Therefore we also have $\nabla_x D_{\psi^*} = 0$ and $\nabla_x^2 D_{\psi^*} = 0$ for $x \in \text{supp } p_\mathcal{D}$. Using these two conditions, we see that $\nabla_\theta^2 \mathcal{L}(\theta^*, \psi^*) = 0$.

To see Eq.25 and Eq.26, simply take the derivatives of Eq.23 and evaluate at $(\theta^*, \psi^*)$.

**Lemma C.2.4.** *The gradient $\nabla_\psi R_i(\theta, \psi)$ of the regularization terms $R_i$, $i \in \{1, 2\}$, w.r.t. $\psi$ are*

$$\nabla_\psi R_1(\theta, \psi) = \gamma \mathbb{E}_{x \sim p_\mathcal{D}}[\nabla_{\psi, x} D_\psi \nabla_x D_\psi] \tag{28}$$

$$\nabla_\psi R_2(\theta, \psi) = \gamma \mathbb{E}_{x \sim p_\theta}[\nabla_{\psi, x} D_\psi \nabla_x D_\psi] \tag{29}$$

*Proof.* See Mescheder *et al.* [52], Lemma D.3.

**Lemma C.2.5.** *The second derivatives $\nabla_\psi^2 R_i(\theta^*, \psi^*)$ of the regularization terms $R_i$, $i \in \{1, 2\}$, w.r.t. $\psi$ at $(\theta^*, \psi^*)$ are both given by*

$$L_{DD} = \gamma \mathbb{E}_{x \sim p_\mathcal{D}}[AA^\top] \tag{30}$$

*where $A = \nabla_{\psi, x} D_{\psi^*}$. Moreover, both regularization terms satisfy $\nabla_{\theta, \psi} R_i(\theta^*, \psi^*) = 0$.*

*Proof.* See Mescheder *et al.* [52], Lemma D.4.

Given Lemma C.2.3, Lemma C.2.5 and Eq.20, we can now show that the Jacobian of the regularized gradient field at the equilibrium point is given by

$$\mathbf{J}_{\tilde{v}}\Big|_{(\theta^*, \psi^*)} = \begin{pmatrix} 0 & -K_{DG}^\top \\ K_{DG} & M_{DD} \end{pmatrix} \tag{31}$$

where $M_{DD} = K_{DD} - L_{DD}$. To prove our main theorem, we need to examine $\mathbf{J}_{\tilde{v}}$ when restricting it to the space orthogonal to $\mathcal{T}_{(\theta^*, \psi^*)} \mathcal{M}_G \times \mathcal{M}_D$.

**Lemma C.2.6.** *Assume Assumptions II and III hold. If $v \neq 0$ is not in $\mathcal{T}_{\psi^*} \mathcal{M}_D$, then $v^\top M_{DD} v < 0$.*

*Proof.* By Lemma C.2.3 and Lemma C.2.5, we have

$$v^\top K_{DD} v = f''(0)\mathbb{E}_{\substack{x \sim p_\mathcal{D} \\ y \sim p_\mathcal{D}}} \left[((\nabla_\psi D_{\psi^*}(x) - \nabla_\psi D_{\psi^*}(y))^\top v)^2\right] \tag{32}$$

$$v^\top L_{DD} v = \gamma \mathbb{E}_{x \sim p_\mathcal{D}} \left[\|Av\|^2\right] \tag{33}$$

By Assumption II, we have $f''(0) < 0$. Therefore $v^\top M_{DD} v \leq 0$. Suppose $v^\top M_{DD} v = 0$, this implies

$$(\nabla_\psi D_{\psi^*}(x) - \nabla_\psi D_{\psi^*}(y))^\top v = 0 \quad \text{and} \quad Av = 0 \tag{34}$$

for all $(x, y) \in \operatorname{supp} p_{\mathcal{D}} \times \operatorname{supp} p_{\mathcal{D}}$. Recall the definition of $h(\psi)$ from Eq.16. Using the fact that $D_{\psi^*} = C$ and $\nabla_x D_{\psi^*} = 0$ for $x \in \operatorname{supp} p_{\mathcal{D}}$, we see that the Hessian of $h(\psi)$ at $\psi^*$ is

$$\nabla_\psi^2 h(\psi^*) = 2\mathbb{E}_{\substack{x \sim p_{\mathcal{D}} \\ y \sim p_{\mathcal{D}}}}[(\nabla_\psi D_{\psi^*}(x) - \nabla_\psi D_{\psi^*}(y))(\nabla_\psi D_{\psi^*}(x) - \nabla_\psi D_{\psi^*}(y))^\top + AA^\top] \quad (35)$$

The second directional derivative $\partial_v^2 h(\psi)$ is therefore

$$\partial_v^2 h(\psi) = 2\mathbb{E}_{\substack{x \sim p_{\mathcal{D}} \\ y \sim p_{\mathcal{D}}}}\left[\left|(\nabla_\psi D_{\psi^*}(x) - \nabla_\psi D_{\psi^*}(y))^\top v\right|^2 + \|Av\|^2\right] = 0 \quad (36)$$

By Assumption III, this can only hold if $v \in \mathcal{T}_{\psi^*}\mathcal{M}_D$.

**Lemma C.2.7.** *Assume Assumption III holds. If $w \neq 0$ is not in $\mathcal{T}_{\theta^*}\mathcal{M}_G$, then $K_{DG}w \neq 0$.*

*Proof.* See Mescheder *et al.* [52], Lemma D.6.

*Proof for the main theorem.* Given previous lemmas, by choosing local coordinates $\theta(\alpha, \gamma_G)$ and $\psi(\beta, \gamma_D)$ for $\mathcal{M}_G$ and $\mathcal{M}_D$ such that $\theta^* = 0$, $\psi^* = 0$ as well as

$$\mathcal{M}_G = \mathcal{T}_{\theta^*}\mathcal{M}_G = \{0\}^k \times \mathbb{R}^{n-k} \quad (37)$$

$$\mathcal{M}_D = \mathcal{T}_{\psi^*}\mathcal{M}_D = \{0\}^l \times \mathbb{R}^{m-l} \quad (38)$$

our proof is *exactly* the same as Mescheder *et al.* [52], Theorem 4.1.

# D  Hyperparameters, training configurations, and compute

We implement our models on top of the official StyleGAN3 code base. While the loss function and the models are implemented from scratch, we reuse support code from the existing implementation whenever possible. This includes exponential moving average (EMA) of generator weights [28], non-leaky data augmentation [30], and metric evaluation [32].

**Training schedule.**  To speed up the convergence early in training, we specify a cosine schedule for the following hyperparameters before they reach their target values:

- Learning rate
- $\gamma$ for $R_1$ and $R_2$ regularization
- Adam $\beta_2$
- EMA half-life
- Augmentation probability

We call this early training stage the burn-in phase. Burn-in length and schedule for each hyperparameter are listed in Table 9 for each experiment. A schedule for the EMA half-life can already be found in Karras *et al.* [30], albeit they use a linear schedule. A lower initial Adam $\beta_2$ is crucial to the initial large learning rate as it allows the optimizer to adapt to the gradient magnitude change much quicker. We use a large initial $\gamma$ to account for that early in training: $p_\theta$ and $p_\mathcal{D}$ are far apart and a large $\gamma$ smooths both distributions more aggressively which makes learning easier. Augmentation is not necessary until $D$ starts to overfit later on; thus, we set the initial augmentation probability to 0.

**Dataset augmentation.**  We apply horizontal flips and non-leaky augmentation [30] to all datasets where augmentation is enabled. Following [30], we include pixel blitting, geometric transformations, and color transforms in the augmentation pipeline. We additionally include cutout augmentation which works particularly well with our model, although it does not seem to have much effect on StyleGAN2. We also find it beneficial to apply color transforms less often and thus set their probability multiplier to 0.5 while retaining the multiplier 1 for other types of augmentations. As previously mentioned, we apply a fixed cosine schedule to the augmentation probability rather than adjusting it adaptively as in [30]. We did not observe any performance degradation with this simplification.

**Network capacity.**  We keep the capacity distribution for each resolution the same as in [30, 32]. We place two residual blocks per resolution which makes our model roughly $3\times$ as deep, $1.5$–$3\times$ as wide as StyleGAN2 while maintaining the same model size on CIFAR-10 and FFHQ. For the ImageNet model, we double the number of channels which results in roughly $4\times$ as many parameters as the default StyleGAN2 configuration.

**Mixed precision training.**  We apply mixed precision training as in [30, 32] where all parameters are stored in FP32, but cast to lower precision along with the activation maps for the 4 highest resolutions. We notice that using FP16 as the low precision format cripples the training of our model. However, we see no problem when using BFloat16 instead.

**Class conditioning.**  For class conditional models, we follow the same conditioning scheme as in [30]. For $G$, the conditional latent code $z'$ is the concatenation of $z$ and the embedding of the class label $c$, specifically $z' = \text{concat}(z, \text{embed}(c))$. For $D$, we use a projection discriminator [54] which evaluates the dot product of the class embedding and the feature vector $D'(x)$ produced by the last layer of $D$, concretely $D(x) = \text{embed}(c) \cdot D'(x)^\top$. We do not employ any normalization-based conditioning such as AdaIN [29], AdaGN [7, 33], AdaBN [3] or AdaLN [58] for simplicity, even though they improve FID considerably.

**Stacked MNIST.**  We base this model off of the CIFAR-10 model but without class conditioning. We disable all data augmentation and shorten the burn-in phase considerably. We use a constant learning rate and did not observe any benefit of using a lower learning rate later in the training.

**Compute resources.**  We train the Stacked MNIST and CIFAR-10 models on an $8\times$ NVIDIA L40 node. Training took 7 hours for Stacked MNIST and 4 days for CIFAR-10. The FFHQ model was trained on an $8\times$ NVIDIA A6000 f0r roughly 3 weeks. The ImageNet model was trained on NVIDIA A100/H100 clusters and training took one day on 32 H100s (about 5000 H100 hours).

| Hyperparameter | Stacked MNIST | CIFAR-10 | FFHQ | | ImageNet | |
|---|---|---|---|---|---|---|
| Resolution | $32 \times 32$ | $32 \times 32$ | $256 \times 256$ | $64 \times 64$ | $32 \times 32$ | $64 \times 64$ |
| Class conditional | - | ✓ | - | - | ✓ | ✓ |
| Number of GPUs | 8 | 8 | 8 | 8 | 32 | 64 |
| Duration (Mimg) | 10 | 250 | 200 | 100 | 1000 | 1000 |
| Burn-in (Mimg) | 2 | 20 | 20 | 20 | 200 | 200 |
| Minibatch size | 512 | 512 | 256 | 256 | 4096 | 4096 |
| Learning rate | $2 \times 10^{-4}$ | $2 \times 10^{-4} \to 5 \times 10^{-5}$ | $2 \times 10^{-4} \to 5 \times 10^{-5}$ | $2 \times 10^{-4} \to 5 \times 10^{-5}$ | $2 \times 10^{-4} \to 5 \times 10^{-5}$ | $2 \times 10^{-4} \to 5 \times 10^{-5}$ |
| $\gamma$ for $R_1$ and $R_2$ | $1 \to 0.1$ | $0.05 \to 0.005$ | $150 \to 15$ | $2 \to 0.2$ | $0.5 \to 0.05$ | $1 \to 0.1$ |
| Adam $\beta_2$ | $0.9 \to 0.99$ | $0.9 \to 0.99$ | $0.9 \to 0.99$ | $0.9 \to 0.99$ | $0.9 \to 0.99$ | $0.9 \to 0.99$ |
| EMA half-life (Mimg) | $0 \to 0.5$ | $0 \to 5$ | $0 \to 0.5$ | $0 \to 0.5$ | $0 \to 50$ | $0 \to 50$ |
| Channels per resolution | 768-768-768-768 | 768-768-768-768 | 96-192-384-768-768-768-768 | 384-768-768-768 | 1536-1536-1536-1536 | 1536-1536-1536-1536-1536 |
| ResBlocks per resolution | 2-2-2-2 | 2-2-2-2 | 2-2-2-2-2-2-2 | 2-2-2-2 | 2-2-2-2 | 2-2-2-2-2 |
| Groups per resolution | 96-96-96-96 | 96-96-96-96 | 12-24-48-96-96-96-96 | 48-96-96-96 | 96-96-96-96 | 96-96-96-96-96 |
| $G$ params | 20.73M | 20.78M | 23.06M | 22.43M | 82.91M | 103.57M |
| $D$ params | 20.68M | 21.28M | 23.01M | 22.38M | 86.55M | 107.21M |
| Dataset $x$-flips | - | ✓ | ✓ | ✓ | ✓ | ✓ |
| Augment probability | - | $0 \to 0.55$ | $0 \to 0.3$ | $0 \to 0.3$ | $0 \to 0.5$ | $0 \to 0.4$ |

Table 9: Hyperparameters for each experiment. The decay factor $\beta$ of EMA can be obtained using the formula $\beta = 0.5^{\frac{\text{Minibatch size}}{\text{EMA half-life}}}$, *e.g.* for CIFAR-10, EMA $\beta = 0.5^{\frac{512}{5 \times 10^6}} \approx 0.9999$.

# E   Negative Results and Future Work

Following the convention of Brock *et al.* [3], we report alternative design choices that did not make to our final model. Either because they failed to produce any quantitative improvement or because they would considerably complicate our minimalist design which might be better suited for future study.

- We tried to apply GELU [18], Swish [61], and SMU [2] to $G$ and $D$ and found that doing so deteriorates FID considerably. We did not try on $G$ xor $D$. We posit two independent factors:
  - ConvNeXt in general does not benefit much from GELU (and possibly similar activations). Table 10 and Table 11 in [48]: replacing ReLU with GELU gives little performance gain to ConvNeXt-T and virtually no performance gain to ConvNeXt-B.
  - In the context of GANs, GELU and Swish have the same problem as ReLU: that they have little gradient in the negative interval. Since G is updated from the gradient of D, having these activation functions in D could sparsify the gradient of D and as a result G will not receive as much useful information from D compared to using leaky ReLU.

  This does not explain the strange case of SMU [2]: SMU is a smooth approximation of leaky ReLU and does not have the sparse gradient problem. It is unclear why it also underperformed and future work awaits.

- We tried adding group normalization [92] to $G$ and $D$ and it did not improve FID or training stability. We do not claim that all forms of normalizations are harmful. Our claim in principle c) only extends to normalization layers that explicitly standardizes the mean and standard deviation of the activation maps. This has been verified by prior studies [31, 34, 65]. The harm of normalization layers extends to the adjacent field of image restoration [43, 88]. To the best of our knowledge, EDM2 [34] is currently the strongest diffusion UNet and it does not use normalization layers. However, it does apply normalization to the trainable weights and this improves performance considerably. We expect that applying the normalization techniques in EDM2 would improve our model's performance.

- We tried removing the activation function after the $3 \times 3$ grouped convolution in each residual block as modern architectures [48, 97] typically do not apply non-linearity after depthwise convolution. This worsened FID performance.

- We tried Pixel-Shuffle/Unshuffle [71] for changing the resolution of the activation maps, and found that without low-pass filtering, this led to high frequency artifacts similar to checkerboard artifacts even though Pixel-Shuffle does not have the uneven overlap problem that transposed convolution does. Note that bilinear resampling is equivalent to applying channel duplication/averaging with Pixel-Shuffle/Unshuffle in conjunction with a [1, 2, 1] low-pass kernel. It might be interesting in future studies to explore inplace resampling filters that apply a low-pass filtered Pixel-Shuffle/Unshuffle operation on top of a learned function that changes the number of channels.

- We tried scaling up our model size. We found that allocating more model capacity to lower resolution stages generally did not improve FID, but contributed to more rapid overfitting. Increasing model capacity at higher resolution stages always improves FID in our experiments, however scaling up higher resolution stages is very computationally expensive. Capacity distribution for each resolution stage of the model might be an important topic to explore in future studies.

- For model simplicity, we did not conduct any experiment with a transformer architecture or attention mechanism in general. We are interested to see whether adding attention blocks to a convolutional network (similar to BigGAN [3] and diffusion UNet [20, 33, 34]) or using a pure transformer architecture (similar to DiT [58]) will result in stronger performance. Given the impressive results of EDM2 [34] (which uses UNet), it seems the argument has not yet settled for generative modeling.

- We experimented with Adam $\beta_2 = 0.999$ following common practice in supervised learning and diffusion models, and found that doing so led to stability issues on our ImageNet models. We expect that introducing proper normalization to our model will resolve this problem.

- We tried mixed precision training with IEEE FP16 as this is the low precision format used in StyleGAN2-ADA [30], StyleGAN3 [32], and EDM2 [34]. This crippled the training of

our model and switching to BFloat16 fixed the problem. We expect that introducing proper normalization to our model will allow us to use IEEE FP16 which offers more precision than BFloat16.

- We tried lazy regularization [31] in our early experiments where $R_1$ and $R_2$ were applied once every 8 minibatches. This led to slightly worse FID performance on real world datasets like FFHQ and CIFAR-10. However, it resulted in complete convergence failure on Stacked MNIST and several two dimensional toy datasets (line, circle, 25 Gaussians, *etc*.), indicating potential concerns regarding the mathematical legitimacy of this trick.

# F Qualitative Results

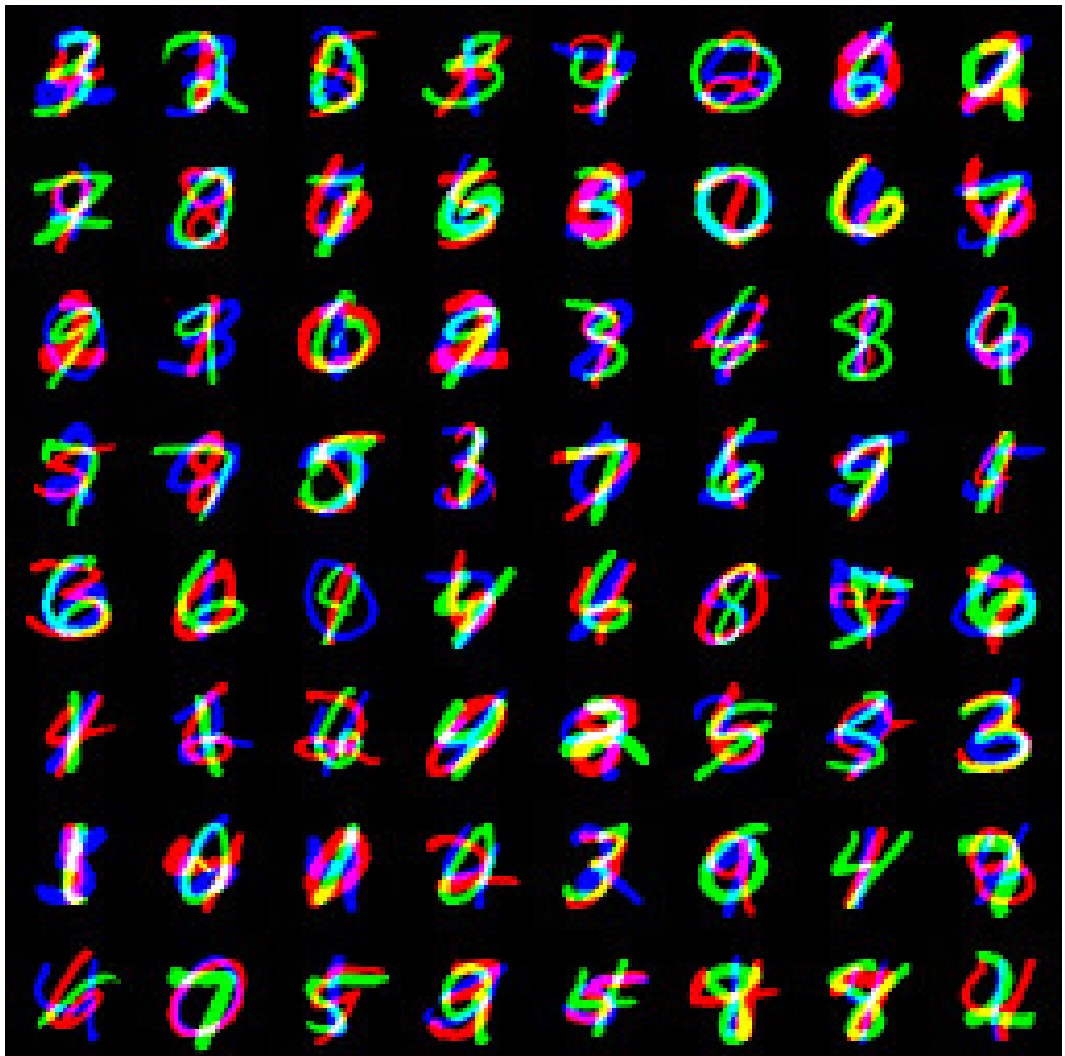

Figure 5: Qualitative examples of sample generation from our Config E on Stacked-MNIST.

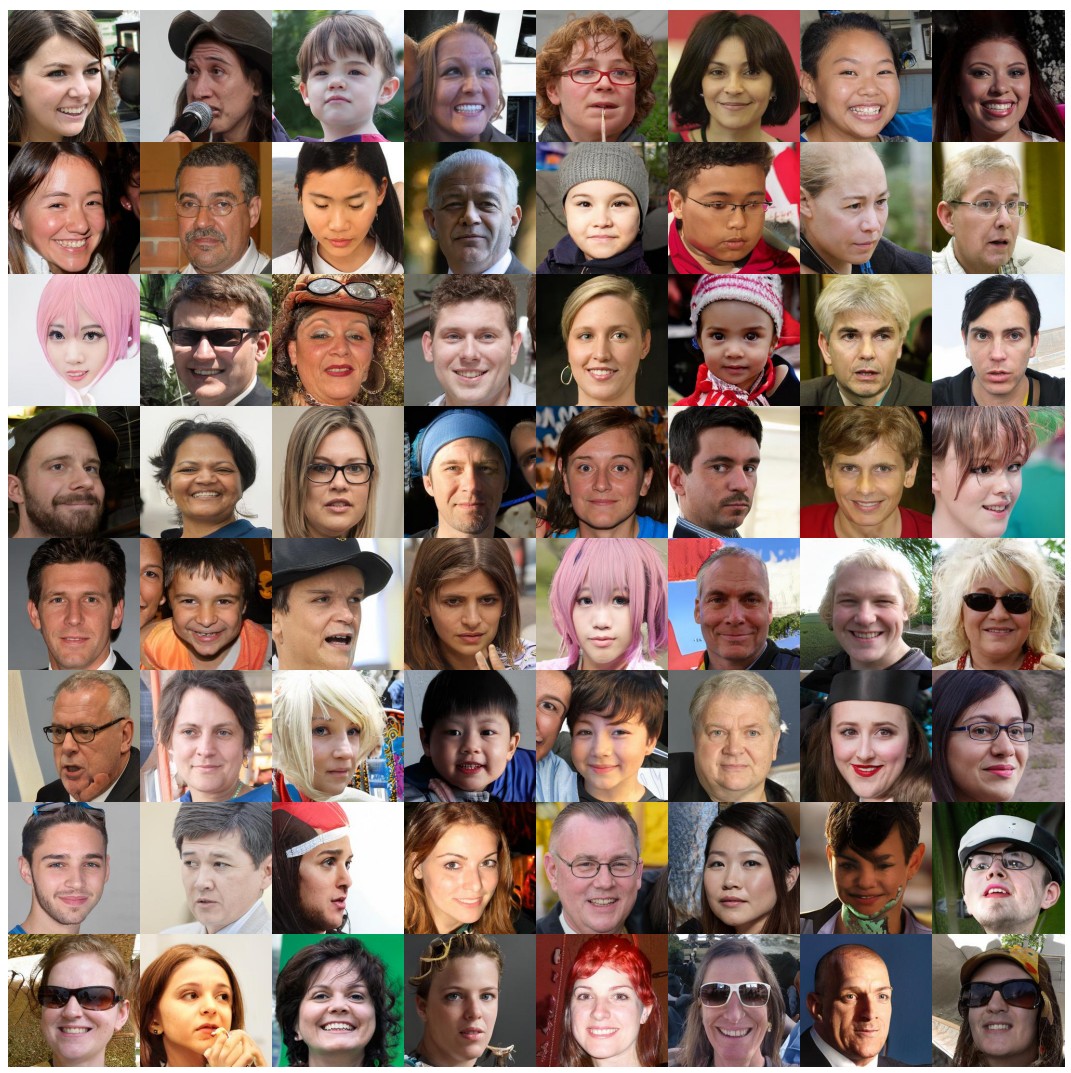

Figure 6: More qualitative examples of sample generation from our Config E on FFHQ-256.

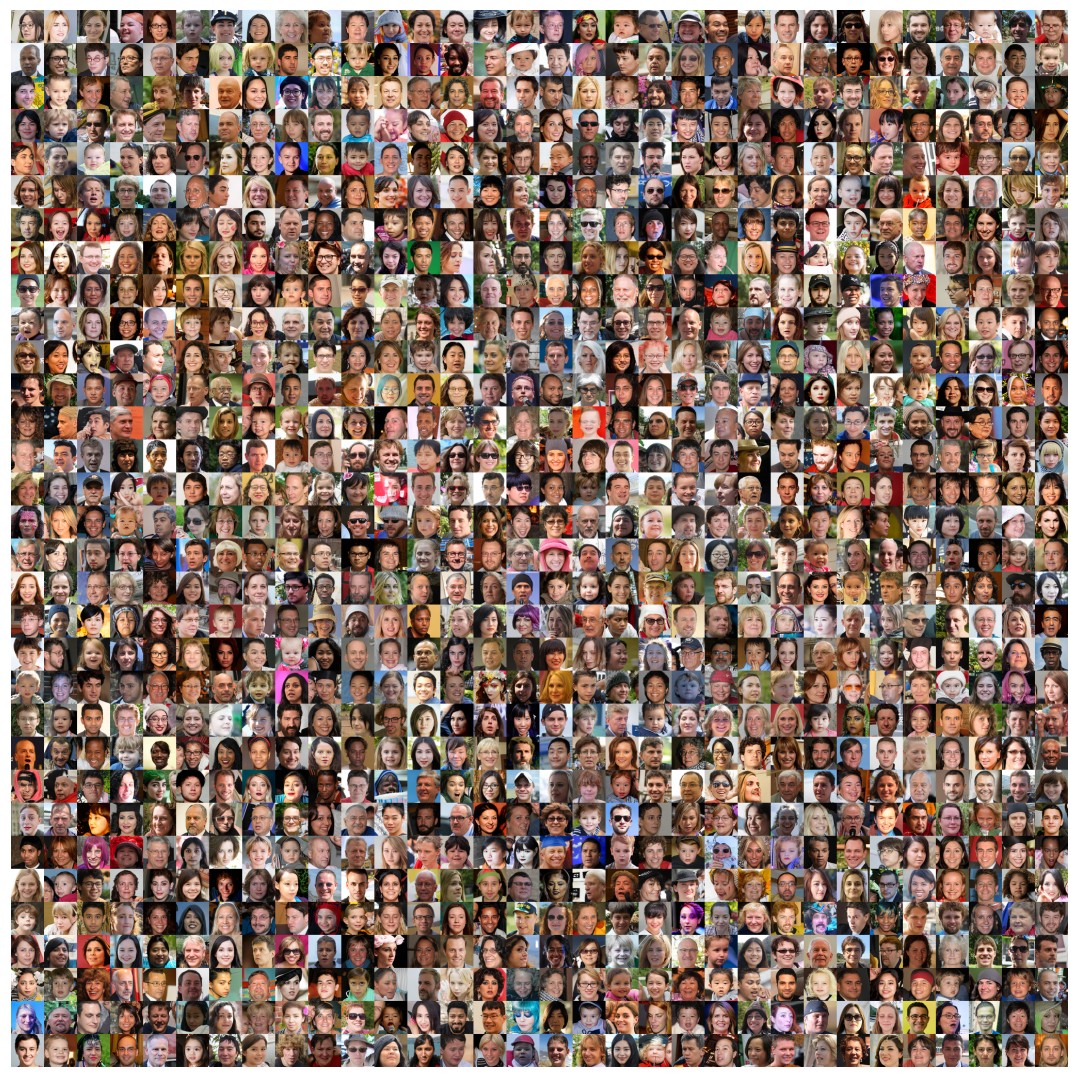

Figure 7: Qualitative examples of sample generation from our Config E on FFHQ-64.

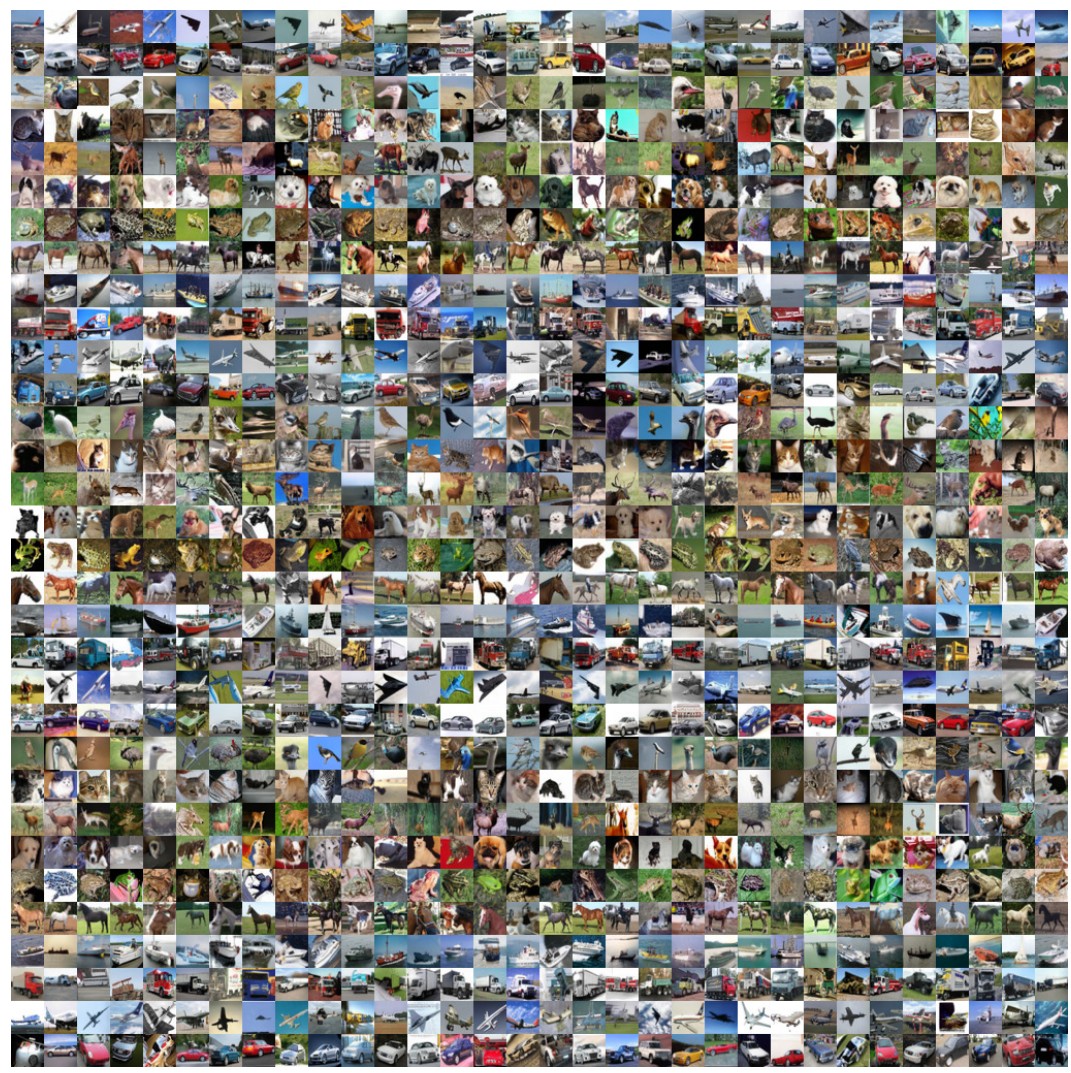

Figure 8: Qualitative examples of sample generation from our Config E on CIFAR-10.

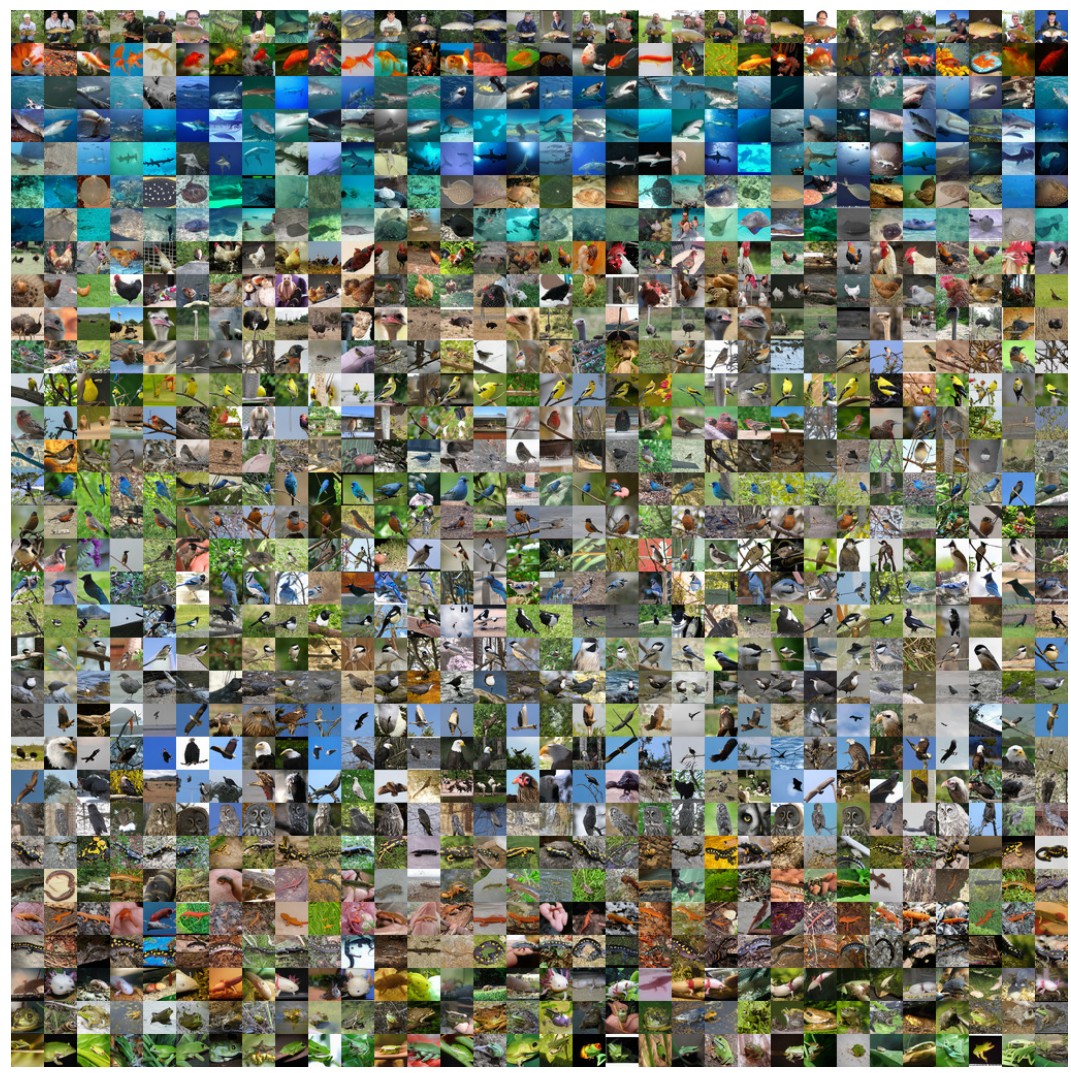

Figure 9: Qualitative examples of sample generation from our Config E on ImageNet-32.

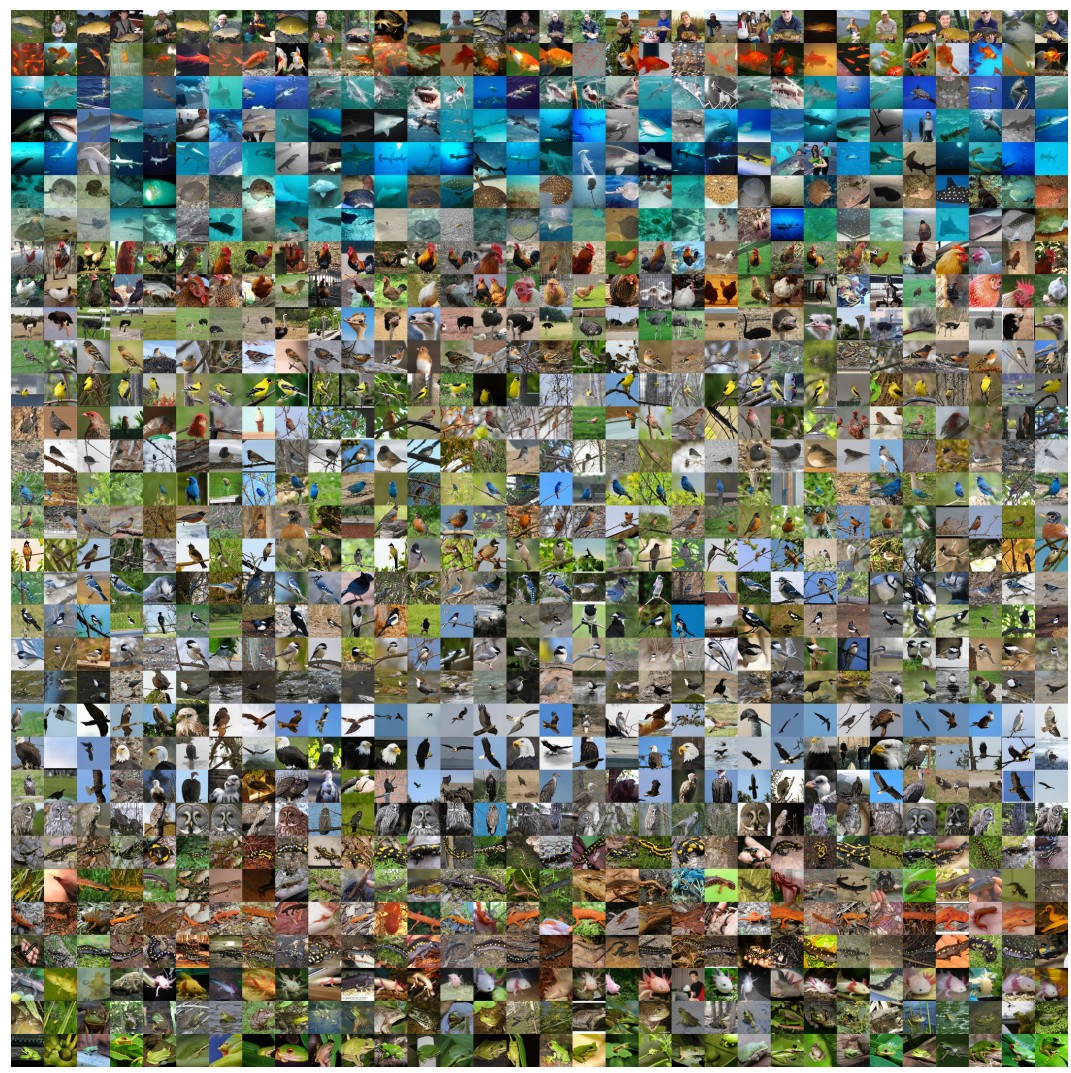

Figure 10: Qualitative examples of sample generation from our Config E on ImageNet-64.

# G Training Curves

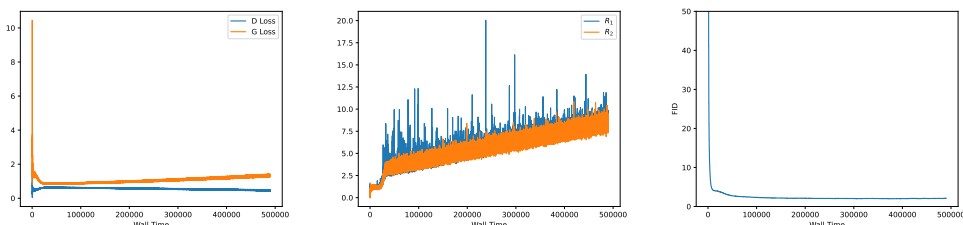

Figure 11: CIFAR-10 training curves.

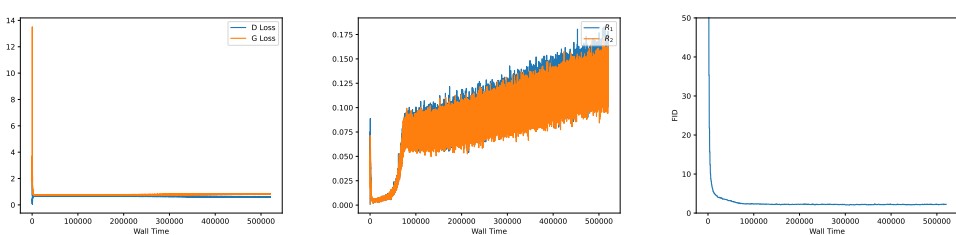

Figure 12: FFHQ-64 training curves.

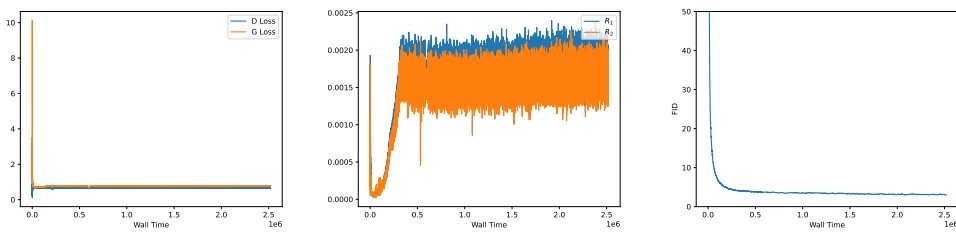

Figure 13: FFHQ-256 training curves.

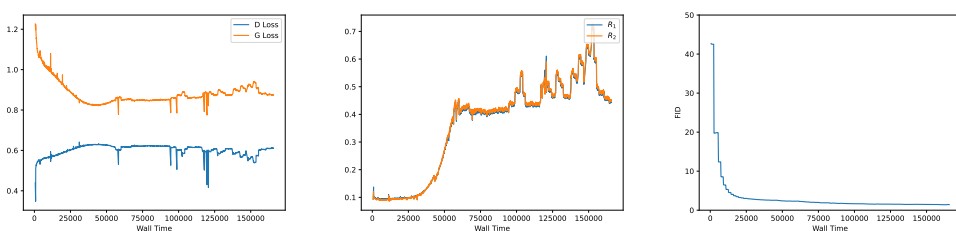

Figure 14: ImageNet-32 training curves.

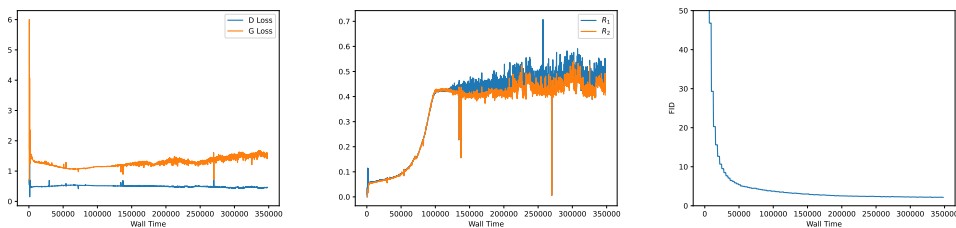

Figure 15: ImageNet-64 training curves.

