# OpenReview forum: "The GAN is dead; long live the GAN! A Modern GAN Baseline"
_NeurIPS.cc/2024/Conference — NeurIPS 2024 poster_

### Official Review · Reviewer_YF9j · 2024-06-19

**Soundness:** 4
**Presentation:** 4
**Contribution:** 4
**Rating:** 9
**Confidence:** 5

**Summary:**

A lot of GANs papers have shrunk in quality since diffusion arose as a strong method. The authors goes against the grain by focusing on modernizing GAN baseline in a principled manner and in doing so they obtain stable, convergent, diversity, and high-quality images comparable to diffusion models. The authors highlight the fact that a lot of the GAN literature is riled with empirical tricks with no theory to back them. They explain that StyleGAN without tricks is very close to the original DCGAN of 2015. Meanwhile a lot more progress as been done in diffusion models leveraging many architectural improves.

They instead start with a Relativistic pairing GAN baseline which, unlike regular GANs, have been shown to have no local-minima basins where the loss must worsen before reaching the true global minimum. However, they show that, just like regular GANs, RpGANs are not guaranteed to converge to the global minimum. They show that, just like regular GANs, gradient penalty on real data (R1) or fake data (R2) guarantee convergence. They suggests using both together. They then show empirically that on StackMNIST, GANs with R1 alone fails, but having both R1, R2, and RpGAN lead to perfect mode coverage and strong stability. As further justification, they also leverage the fact that from a constructive approach, when deriving a maximum-margin GAN, one obtains a gradient penalty with both R1, R2.

Then they use a very careful ablation approach by starting from StyleGAN2, removing all the complicated empirical tricks but keeping R1 penalty, which reduce performance a bit, then moving to RpGAN, adding R2 penalty, then modernizing the architecture, reaching slightly better performance at similar number of parameters, but without tricks. The describe extensively and carefully the effect of different architecture choices on performance.

Experiments on CIFAR-10, Imagenet-32, and FFHQ-256 are shown were the method proposed shows extremely low FID, better than diffusion models and do so while keeping the parameter count low. Results are very good, but work is still needed to scale these methods to text-to-image settings. However, the current results are very promising and the method has been made very clean and carefully designed (similar to ConvNext).

**Strengths:**

Experiments on CIFAR-10, Imagenet-32, and FFHQ-256 are shown were the method proposed shows extremely low FID, better than diffusion models and do so while keeping the parameter count low. The derivation of their architecture is done carefully in a thorough manner.

**Weaknesses:**

Great results on image datasets, but it still need to be scaled to large-scale text-to-image setting.
Many GPUs are used, even for CIFAR-10; this is also in line with diffusion models, but it seems a lot, are all these GPUs needed for CIFAR-10?
Although cleaner, the new baseline still contains a lot of complicated hyperparameter tuning, which change from one dataset to another (in Table 7). It would be great to clean this set of hyperparameters, I suggested some potential ideas in the "questions".

**Questions:**

- Any intuition to why worse performance with GELU and swish activations since these are very successful with transformers? Is this true for both G and D?
- Have you consider a transformer architecture as alternative to convnext-style architecture?
- You mention that normalization can harm GANs, but groupnorm is used in diffusion models and layernorm is used in language models, could this be useful for your approach? Has it been tried before? I saw the references you mentioned to say that normalization is bad for GANs, but I haven't looked at them individually.
- You mention that PresGAN and DDGANs obtain good performances, do you think that some of their ideas could be useful for improve the GAN baseline?
- What is EMA half-life? Is the EMA .999?
- Why do you schedule Adam B2? This seems very atypical.
- Assume $\alpha$ is the lr. Rather than schedule the R1/R2 $\gamma$ manually, why not make it $\alpha\gamma$ so that by decreasing the lr $\alpha$ over time, $\gamma$ is also reduces automatically (thus reducing the hyperparameter tuning). The same could be done with Beta2, by doing $1-((\alpha/alpha_{init})*(1-beta2))$, so moving the 1-.9 = 0.1 to 0.01 (1-.99) when decreasing the lr to 1/10 of its value over training time. I just feel like choices like this would greatly reduce the arbitrary hyperparameter tuning used here which is quite complex in Table 7.

**Limitations:**

The authors have adequately addressed the limitations.

---

> ### Author Rebuttal · Authors · 2024-08-06
>
> Thank you for your feedback.
>
> > “... still need to be scaled to large-scale text-to-image setting.”
>
> For scaling, we are currently running ImageNet-64 experiments to include in our paper; please see our discussion in response to Reviewer 2Hzz. In general, we hope our work is a meaningful first step and that future works can better assess its scaling.
>
> > “Many GPUs are used, even for CIFAR-10; this is also in line with diffusion models, but it seems a lot, are all these GPUs needed for CIFAR-10?”
>
> Using 8 GPUs to train CIFAR-10 is not required; our models can be trained with fewer GPUs at the cost of slower training speed. But, it is becoming a common practice, e.g., for diffusion models; Karras et al. used 8 GPUs to train their CIFAR-10 models in EDM.
>
> > “Any intuition to why worse performance with GELU and swish activations since these are very successful with transformers? Is this true for both G and D?”
>
> In experiments, we tried to apply GELU/Swish/SMU to G and D and found that doing so deteriorates FID considerably. We did not try on G xor D. We posit two independent factors:
> - ConvNeXt in general does not benefit much from GELU (and possibly similar activations). Table 10 and Table 11 in https://arxiv.org/abs/2201.03545: replacing ReLU with GELU gives little performance gain to ConvNeXt-T and virtually no performance gain to ConvNeXt-B.
> - In the context of GANs, GELU and Swish have the same problem as ReLU: that they have little gradient in the negative interval. Since G is updated from the gradient of D, having these activation functions in D could sparsify the gradient of D and as a result G will not receive as much useful information from D compared to using leaky ReLU.
>
> This does not explain the strange case of SMU (https://arxiv.org/abs/2111.04682): SMU is a smooth approximation of leaky ReLU and does not have the sparse gradient problem. It is unclear why it also underperformed and future work awaits. We are happy to add this discussion to supplemental.
>
> > “Have you considered a transformer architecture?”
>
> We have not conducted any experiments, but are also interested to see how they perform. In particular, whether adding attention blocks to a ConvNet (similar to BigGAN and diffusion UNet) or using a pure transformer architecture (similar to DiT) will result in stronger performance. Given the impressive results of EDM2 (which uses UNet), it seems the argument has not yet settled for generative modeling.
>
> > “You mention that normalization can harm GANs, but groupnorm is used in diffusion models and layernorm is used in language models, could this be useful for your approach? Has it been tried before? I saw the references you mentioned to say that normalization is bad for GANs.”
>
> We tried adding groupnorm to G and D and it did not improve FID or training stability. We would like to clarify that we do not intend to claim that all forms of normalizations are harmful. Our claim only extends to normalization layers that explicitly standardizes the mean and stddev of the activation maps. This has been verified by prior studies:
> - Instance norm harms GANs: https://arxiv.org/abs/1912.04958
> - Group norm harms diffusion models: https://arxiv.org/abs/2312.02696
> - Group norm harms VAEs: https://arxiv.org/abs/2405.14477, pg. 5
>
> The harm of normalization layers extends to the adjacent field of image restoration (https://arxiv.org/abs/1707.02921, sec 3.1; https://arxiv.org/abs/1809.00219, supplemental sec 1). To the best of our knowledge, EDM2 is currently the strongest diffusion UNet and it does not use normalization layers. However, it does apply normalization to the trainable weights and this improves performance considerably. We expect that applying the normalization techniques in EDM2 would improve our model's performance.
>
> > “You mention that PresGAN and DDGANs obtain good performances, do you think that some of their ideas could be useful?”
>
> Yes, it is possible. We excluded these ideas as they are not an indispensable part of a minimal GAN framework. DDGAN in particular combines GANs and diffusion models - a promising research direction. Prior work has explored this in various flavors: DDGAN formulates GAN as a fast diffusion solver; Diffusion GAN (https://arxiv.org/abs/2206.02262) applies diffusion to GANs as a non-leaky augmentation; Diffusion2GAN (https://arxiv.org/abs/2405.05967) distills a diffusion model into a GAN; PaGoDA (https://arxiv.org/abs/2405.14822) inverts a pretrained diffusion model to obtain the posterior p(z | x) and facilitates the combination of GANs and MLE.
>
> > “What is EMA half-life? Is the EMA .999?”
>
> We follow the notation of Karras et al. in StyleGAN and EDM. EMA beta can be obtained using the formula ema_beta = 0.5 ** (batch_size / ema_half_life).
>
> > “Why do you schedule Adam B2? This seems very atypical.”
>
> It is atypical. The gradient magnitude of G and D varies drastically early in training. This effect is further amplified when we use a large initial learning rate, leading to large loss oscillations early on. Using a constant Adam beta2 such as 0.99 produces a loss that always stabilizes in a few hundred steps, and the initial oscillations do not seem to have any long-term negative impact.
> However, by annealing Adam beta2, we can reduce the initial loss oscillations: now, we no longer need to wait a few hundred steps for the loss to settle. In particular, a lower initial Adam beta2 allows the optimizer to adapt to large gradient magnitude changes much quicker and so stabilize training earlier. Similar reasoning for tweaking Adam beta2 is given in EDM2 (https://arxiv.org/abs/2312.02696, pg. 18). The benefit of scheduling Adam beta2 is marginal, but since we already schedule multiple hyperparameters, we might as well schedule Adam beta2 too.
>
> > Hyperparameter scheduling (question truncated for space).
>
> We agree that your suggestion will ease the hyperparameter tuning, and we plan to introduce something similar for the automatic configuration of our model.

---

> > ### Comment · Reviewer_YF9j · 2024-08-07
> > **response**
> >
> > Thank you for addressing my questions and concerns. Having some of these short discussions in the appendix would indeed be useful to the reader. The extra imagenet-64 experiment will also help.
> >
> > I will increase my score by 1. This method is particularly good and helps bring GANs extremely close to diffusion performance which is quite important for the field of GANs to regain usefulness. The design choices are quite interesting. This reminds me a bit of the very useful experiments detailed discussions in the BigGAN paper which really helped pave the way to better GANs.

---

### Official Review · Reviewer_2Hzz · 2024-07-10

**Soundness:** 2
**Presentation:** 2
**Contribution:** 2
**Rating:** 5
**Confidence:** 3

**Summary:**

This paper introduces R3GAN as a GAN baseline that simplifies and modernizes the architecture by replacing ad-hoc tricks with modern designs. R3GAN utilized a regularized relativistic GAN loss coupled with zero-centered gradient penalties on both real and generated data, to addresses mode dropping and non-convergence, providing local convergence guarantees. The proposed loss allows for the elimination of unnecessary tricks and the adoption of advanced architectures. R3GAN demonstrates improved stability and competitive performance against state-of-the-art models on multiple datasets such as StackedMNIST, CIFAR, ImageNet-32, and FFHQ-256.

**Strengths:**

+ The writing of the paper is clear, making it easy to understand and easy to follow.
+ One contribution of this paper is the simplification and optimization of the StyleGAN network architecture, which, as demonstrated by experiments, achieves considerable performance without the need for many elaborate tricks.
+ The paper provides theoretical analysis to prove the convergence of the loss used.

**Weaknesses:**

- The novelty of the method is somewhat limited, as both relativistic pairing GAN (RpGAN) and zero-centered gradient penalties (0-GPs) are previously proposed and validated approaches in the field of GANs.
- Small-scale experiments to validate the significant improvements in stability and diversity provided by the used loss are insufficient; more complex datasets should be used to verify its effectiveness. As shown in Table 2, the improvement from config C compared to config B is not significant. Generally, stability can allow the model to converge better, thereby enhancing the final results, and diversity can be measured by metrics such as recall or coverage.

**Questions:**

How scalable is the proposed model? It is suggested to validate the effectiveness of the method on higher-resolution datasets and more complex tasks. This would align with the motivation to serve as a modern baseline, especially in the current context where both model parameter size and training data scale are increasing.

**Limitations:**

The authors have discussed the limitations and potential negative societal impact of their work.

---

> ### Author Rebuttal · Authors · 2024-08-06
>
> Thank you for your feedback.
>
> > “The novelty of the method is somewhat limited, as both relativistic pairing GAN (RpGAN) and zero-centered gradient penalties (0-GPs) are previously proposed and validated approaches in the field of GANs.”
>
> While these components have been proposed separately, none matches our contributions:
> - **New theoretical and empirical contribution**: 0-GPs have been proposed in the context of classic GANs, but nobody has studied how they interact with the training dynamics of RpGANs. We provide both theoretical and empirical evidence on how they address the non-convergence of RpGANs, which ultimately allows us to derive a well-behaved GAN loss that is resilient to GAN optimization problems.
> - **New practical contribution**: We establish a clean and cohesive GAN framework with just the must-have components. With the new loss, we show that a simple non-StyleGAN based architecture can beat StyleGAN2 by a large margin.
>
> > “How scalable is the proposed model? It is suggested to validate the effectiveness of the method on higher-resolution datasets and more complex tasks. This would align with the motivation to serve as a modern baseline, especially in the current context where both model parameter size and training data scale are increasing.”
>
> We present early evidence: To complement our ImageNet-32 result, we are currently running ImageNet-64 experiments by stacking another stage onto our ImageNet-32 architecture. Note that, to evaluate EDM, Karras et al. (2022) used ImageNet-64 as their biggest dataset. Without hyperparameter tuning, we achieved FID 2.56, which is in between ADM (FID: 2.91, 250 DDIM steps) and EDM (FID: 2.23, 79 steps with EDM sampler). We achieve this without techniques like adaGN and attention in ADM/EDM. Our result uses ~90M parameters while EDM and ADM use ~300M. This is evidence that our model is more efficient at its current scale and will hopefully scale well.
>
> For more complex tasks, we are unsure what the reviewer would like to see - please suggest something. To the best of our knowledge, ImageNet is the one of the largest and best studied closed-world datasets. If the reviewer meant extending the setting to open-world text-to-image generation, then this requires significantly more compute (and monetary cost!). It also may require additional techniques like text encoders, self/cross-attention, and conditional modulation.
>
> More generally, addressing scalability is secondary to our top priority of addressing the GAN “myths” that led to the shrinking (as YF9j puts it) of GAN research: GANs are difficult to train, GANs do not work without many tricks, GANs are fragile and do not work with powerful modern vision backbones. Scalability, while also highly important to GAN applications, is dependent upon proving the feasibility of a clean GAN framework, via a minimal model that beats prior work. Of course, we agree with the reviewer on the practical importance of scalability. We are not OpenAI or Meta with thousands of GPUs, but our added experiment will provide evidence for the scalability of our model given our compute constraints.
>
> Finally, we will open source our code and pretrained models and we welcome the community to test the scalability of our model.
>
> > "Small-scale experiments to validate the significant improvements in stability and diversity provided by the used loss are insufficient; more complex datasets should be used to verify its effectiveness."
>
> We acknowledge that stability and diversity are only directly validated on StackedMNIST (Figure 1 / Table 1). We have added diversity via the recall metric for all other experiments below. Concerning stability, we propose to add convergence plots for all datasets and for loss-ablated models, similar to Figure 1, in supplemental material, with discussion in the main paper.
>
> That said, StackedMNIST itself should not be dismissed so easily: it is a challenging case for stability because of the limited data and that data having an all-black background. This makes D's job to reject fake samples extremely easy early on, causing instability. FFHQ is generally more stable and the original StyleGAN loss is likely to work in terms of stability, but we show that our approach gives better results in terms of FID.
>
> > [Following on; concerning the impact of stability on performance] “As shown in Table 2, the improvement from config C [well-behaved loss] compared to config B [no well-behaved loss] is not significant.”
>
> Improving performance may need both a well-behaved loss _and_ a more powerful backbone, as is the case here. Both Config-B and C have a weak DCGAN-style backbone. Assuming convergence is stable, G and D must still be sufficiently powerful for a given dataset, otherwise performance will saturate regardless of the loss used. With a weak backbone, it is not surprising that replacing the loss does not result in a drastic improvement. With a stronger backbone in D and E, performance increases. Adding convergence plots for this more complex FFHQ data will contextualize this result with respect to stability.
>
> > “... and diversity can be measured by metrics such as recall …”
>
> Thanks - our oversight - we report recall numbers below.
>
> On ImageNet-32, we obtain a recall of 0.63, comparable to ADM (recall 0.63).
>
> On FFHQ-256, after gamma hyperparameter tuning, we achieved a lower FID of 2.77 and a recall of 0.50, outperforming StyleGAN2 (FID: 3.78, recall: 0.43).
>
> On CIFAR-10, we achieved a slightly better FID of 1.97 and recall 0.57 after gamma hyperparameter tuning. By comparison, StyleGAN-XL has a slightly lower FID of 1.85 but a much worse recall of 0.47. StyleGAN2-Ada has a considerably worse FID of 2.42 but a slightly higher recall of 0.59. We attribute the recall gap to hyperparameter tuning, as we tune hyperparameters for best FID performance.
>
> For ablations, Config-C, using our new loss, achieved recall 0.24, and Config-B with the original StyleGAN loss achieved a worse recall of 0.22.

---

> > ### Comment · Reviewer_2Hzz · 2024-08-14
> >
> > I appreciate the author’s response. Considering that the authors reply to my questions and addressed some of my concerns, I decided to increase my score to Borderline Accept. However, I still believe that conducting experiments on higher resolution datasets (such as ImageNet-128, ImageNet-256, ImageNet-512) would significantly enhance the contribution of this paper to the GAN research field.

---

### Official Review · Reviewer_otaU · 2024-07-13

**Soundness:** 3
**Presentation:** 3
**Contribution:** 3
**Rating:** 6
**Confidence:** 3

**Summary:**

The authors posit that the main reason GAN research has been slow in recent years is due to the most foundational StyleGAN2 not having undergone major architectural changes, essentially due to a lack of convergence guarantee in GAN objectives and being prone to mode collapse. This has limited the scope of architectural changes and further GAN research.
Two identifiable goals of the paper are to mathematically derive a well-behaved regularized loss which guarantees GAN convergence and diversity, provide empirical evidence for the same, and now being free to make unconstrained design choices attempt to develop a GAN baseline which inherits features from modern vision architectures.

The authors' methodology is inspired by two key works: (i) RpGAN [20], which addressed mode dropping in GANs with a better loss landscape, theoretically confirmed by Sun et al. [65] to contain only global minima; and (ii) Mescheder et al. [43], which provided evidence for local convergence with zero-centered gradient penalties in GAN objectives. By extending zero-centered gradient penalties to the RpGAN formulation, the authors address both mode dropping and unstable training. They offer theoretical proof for local convergence and present experimental evidence supporting global convergence, though no theoretical proof for the latter is provided.
Design changes in Config D are motivated by ResNet [14] and ConvNeXt [41], selectively applying aspects consistent with the authors' observations. Following findings in [31, 29], they eliminate normalization in StyleGAN in B and introduce fix-up initialization [77] in D to prevent variance explosion. Changes in E are inspired by ResNeXt [74], incorporating group convolutions to increase bottleneck width and followed by bottleneck inversion for efficient use. This simple baseline is empirically found to outperform previous GAN methods and some recent diffusion based models on several unconditional and conditional generation tasks.

**Strengths:**

1. The paper presents a combination of well-known techniques by extending the RpGAN formulation with zero-centered gradient penalties. 2. While the individual components like RpGAN, zero-centered gradient penalties, and modern vision architectures (ResNet, ConvNeXt) are not new, their integration into a cohesive GAN framework is a notable contribution.
3. The submission is technically sound with well-supported claims through theoretical analysis and experimental results.
4. The paper is clearly written and well-organized.
5. The results are significant, with improved performance over existing GANs and some diffusion models on various datasets.

**Weaknesses:**

1. While experimental results are well presented, there seems to be a lack of information about the training setup, hyperparameters used, number of inference steps, etc pertaining to the diffusion based models in Tables 4, 5 and 6. It would be useful to include this information in the supplementary or the tables itself for a transparent comparison.

**Questions:**

1. To clarify, on line 132 on page 4, “857” corresponds to StyleGAN backbone trained on WGAN-GP loss and “881” corresponds to the mini-batch standard deviation trick added to the same? Would be better to clarify the exact configuration and include the two cases in table 1 and figure 1.

**Limitations:**

-

---

> ### Author Rebuttal · Authors · 2024-08-06
>
> Thank you for your feedback.
>
> > “there seems to be a lack of information about the training setup, hyperparameters used, number of inference steps, etc pertaining to the diffusion based models in Tables 4, 5 and 6.”
>
> The diffusion model numbers in these tables are directly taken from reports in existing papers. Below, we have added the commonly-reported number of function evaluations (NFEs) for each. For the camera ready, we will add more detailed training setup descriptions and a hyperparameter table.
>
> Table 4:
> - LDM: from https://arxiv.org/abs/2112.10752, Table 18, NFE: 200
> - ADM (DDIM & DPM-Solver) & Diffusion Autoencoder: from https://arxiv.org/abs/2312.06285, Table 1, NFE: 500
>
> Table 5:
> - DDPM: from https://arxiv.org/abs/2006.11239, Table 1, NFE: 1000
> - DDIM: from https://arxiv.org/abs/2010.02502, Table 1, NFE: 50
> - VE & VP: from https://arxiv.org/abs/2206.00364, Table 2, NFE: 35
>
> Table 6:
> - ADM  -> https://arxiv.org/pdf/2301.11706 Table 3, NFE: 1000
> - DDPM-IP> https://arxiv.org/pdf/2301.11706 Table 3, NFE 1000
> - VDM -> https://proceedings.neurips.cc/paper_files/paper/2021/file/b578f2a52a0229873fefc2a4b0 -> NFE 1000
> - DDPM++ -> https://openreview.net/forum?id=PxTIG12RRHS -> NFE 1000
>
> > “To clarify, on line 132 on page 4, “857” corresponds to StyleGAN backbone trained on WGAN-GP loss and “881” corresponds to the mini-batch standard deviation trick added to the same? Would be better to clarify the exact configuration…”
>
> Not quite. Although the mini-batch standard deviation trick was popularized by StyleGAN, it was introduced in Progressive GAN (https://arxiv.org/abs/1710.10196). Thus, we took these numbers from Table 4 of the Progressive GAN paper. “857” corresponds to a low-capacity version of the progressive GAN trained with WGAN-GP loss and “881” adds the minibatch stddev trick. We will clarify the reference in the camera-ready paper.
>
> > “... and include the two cases in table 1 and figure 1.”
>
> While in principle this is a good idea to contextualize the gain, in practice it is a little tricky: ProGAN/StyleGAN are quite different models, and Table/Figure 1 aim to show without complication the effect of different losses on our small simple model. We cannot ‘strip’ ProGAN/StyleGAN meaningfully, and so their performance may be better than what our simple model achieves. We do not want to confuse the reader by adding new lines to the plots that are incomparable, obfuscating the point we are trying to make about stability. If the reviewer has other ideas for how to make this comparison, we welcome them.

---

> > ### Comment · Reviewer_otaU · 2024-08-14
> >
> > Thanks for your response. After going through the response and other response to other reviewers, I have decided to keep my current ratings.

---

### Author Rebuttal · Authors · 2024-08-05

Thank you everyone for your constructive feedback.

In summary, all reviewers found that the paper had strengths:
- The paper is clearly written.
- The claims are well supported with both theoretical and empirical evidence.
- The theoretical insights allow a method that is simpler than past GAN works with fewer tricks.
- The results are significant, being better than many existing GANs and some diffusion methods.

However, there are some weaknesses and questions:
- Novelty: Reviewer 2Hzz believes that novelty is somewhat limited as both RpGAN and zero-centered gradient penalty works both exist; reviewer otaU notes that this is true but that "their integration into a cohesive GAN framework is a notable contribution."
- Technical Retails: Reviewer otaU requests training details for comparison methods and a technical clarification, and reviewer YF9j poses numerous small questions to help us improve our technical explanation. We answer each of these questions.
- Stability/Diversity Evaluation: Reviewer 2Hzz states that "small-scale experiments to validate the significant improvements in stability and diversity ... are insufficient; more complex datasets should be used." We report the recall metric for all datasets in this rebuttal to evaluate diversity, and propose plots to show stability on our more complex datasets.
- Scalability: Reviewer 2Hzz asks "How scalable is the proposed model?" Reviewer YF9j notes this issue too - "work is still needed to scale these methods to text-to-image settings" - but is satisfied to support the work as the "current results are very promising." To provide evidence for scalability, we report preliminary results on ImageNet-64 to contrast to our ImageNet-32 results for this task. But, we defer more significant scalability evaluation to future work - we will open source our code and pretrained models to make this easier.

We hope to answer these questions in detail in response to each individual reviewer, and we look forward to further discussion with you all.

---

### Decision · Program_Chairs · 2024-09-25

**Decision:**

Accept (poster)

**Comment:**

The reviews are overall positive and recognize the solid theoretical foundation, empirical performance, and contribution to modernizing GAN architectures. While there are some concerns about novelty, scalability, and the transparency of the experimental setup, the reviewers believe that it's a valuable contribution to the community given the sole recent focus on diffusion models. I urge the authors to incorporate the feedback in the manuscript.